# Shared behavioral mechanisms underlie *C. elegans* aggregation and swarming

**Siyu Serena Ding[1,2†], Linus J Schumacher[3,4†‡], Avelino E Javer[1,2§#], Robert G Endres[3]\*, André EX Brown[1,2]\***

[1]Instititue of Clinical Sciences, Imperial College London, London, United Kingdom; [2]MRC London Institute of Medical Sciences, London, United Kingdom; [3]Department of Life Sciences, Imperial College London, London, United Kingdom; [4]MRC Centre for Regenerative Medicine, University of Edinburgh, Edinburgh, United Kingdom

**\*For correspondence:**
r.endres@imperial.ac.uk (RGE);
andre.brown@imperial.ac.uk (AEB)

[†]These authors contributed equally to this work

**Present address:** [‡]MRC Centre for Regenerative Medicine, University of Edinburgh, Edinburgh, United Kingdom; [§]Institute of Biomedical Engineering, University of Oxford, Oxford, United Kingdom; [#]Data Science Institute, University of Oxford, Oxford, United Kingdom

**Competing interests:** The authors declare that no competing interests exist.

**Abstract** In complex biological systems, simple individual-level behavioral rules can give rise to emergent group-level behavior. While collective behavior has been well studied in cells and larger organisms, the mesoscopic scale is less understood, as it is unclear which sensory inputs and physical processes matter *a priori*. Here, we investigate collective feeding in the roundworm *C. elegans* at this intermediate scale, using quantitative phenotyping and agent-based modeling to identify behavioral rules underlying both aggregation and swarming—a dynamic phenotype only observed at longer timescales. Using fluorescence multi-worm tracking, we quantify aggregation in terms of individual dynamics and population-level statistics. Then we use agent-based simulations and approximate Bayesian inference to identify three key behavioral rules for aggregation: cluster-edge reversals, a density-dependent switch between crawling speeds, and taxis towards neighboring worms. Our simulations suggest that swarming is simply driven by local food depletion but otherwise employs the same behavioral mechanisms as the initial aggregation.
DOI: https://doi.org/10.7554/eLife.43318.001

## Introduction

Collective behavior has been widely studied in living and non-living systems. While very different in their details, shared principles have begun to emerge, such as the importance of alignment for flocking behavior in both theoretical models and birds (*Bialek et al., 2012*; *Pearce et al., 2014*; *Reynolds, 1987*). Until now, the study of collective behavior has mainly focused on cells and active particles at the microscale, controlled by molecule diffusion and direct contact between cells or particles (*Köhler et al., 2011*; *De Palo et al., 2017*; *Peruani et al., 2012*; *Starruß et al., 2012*), and on animals at the macroscale, aided by long-range visual cues (*Bialek et al., 2012*; *Katz et al., 2011*; *Pearce et al., 2014*). Collective behavior at the intermediate mesoscale is less well-studied, as it is unclear what processes to include *a priori*. At the mesoscale, sensory cues and motility may still be limited by the physics of diffusion and low Reynolds numbers, respectively, yet the inclusion of nervous systems allows for increased signal processing and a greater behavioral repertoire. Do the rules governing collective behavior at this intermediate scale resemble those at the micro- or the macroscale, some mixture of both, or are new principles required?

*C. elegans* collective behavior can contribute to bridging this scale gap. Some strains of this 1 mm-long roundworm are known to aggregate into groups on food (*de Bono and Bargmann, 1998*); here we also report an additional dynamic swarming phenotype that occurs over longer time periods. *C. elegans* represents an intermediate scale not only in physical size but also in behavioral complexity—crawling with negligible inertia, limited to touch and chemical sensing, yet possessing a compact nervous system with 302 neurons (*White et al., 1986*) that supports a complex behavioral repertoire (*Hart, 2006*; *Schwarz et al., 2015*). Wild *C. elegans* form clusters on food at ambient

**eLife digest** Anyone who has watched a flock of birds maneuver through the sky has probably wondered how so many animals coordinate their movements. Often, these seemingly complex group behaviors can be explained by a few simple rules that govern the behavior of the individuals in the group. For example, if each bird flies and reacts to its neighbors in a certain way, the whole flock's flight pattern results from these individual choices.

Computer simulations can help researchers to test how individual behaviors contribute to coordinated group movements. Ding, Schumacher et al. have now used a simulation to uncover the rules that control the behavior of small worms called *Caenorhabditis elegans*, which form large groups while feeding on bacteria.

To gather the data needed to form the computer model, Ding, Schumacher et al. genetically engineered *C. elegans* worms to produce fluorescent proteins. The fluorescence allows the movements of the worms to be monitored automatically in time-lapse movies made from a series of microscope images. The movies show that worm clusters move together over a patch of food, consuming it as they go. As the food disappears, the whole worm cluster moves to a new area in search of more food.

The computer simulation that Ding, Schumacher et al. developed to recreate how the clusters move revealed that individual worms in the group interact according to three rules. Firstly, worms slow down when they have more neighbors. Secondly, when a worm leaves its cluster, it will reverse to rejoin the group. And finally, worms will move towards areas with more neighbors.

It is still not known why the *C. elegans* worms form clusters, but understanding how the individuals in the group interact could help future studies to uncover this reason. Many other organisms benefit from forming similar groups, from single celled bacteria to animals such as birds and fish. The results presented by Ding, Schumacher et al. will therefore help researchers to consider whether there are universal rules that control group behavior.

DOI: https://doi.org/10.7554/eLife.43318.002

oxygen concentrations, as do loss-of-function neuropeptide receptor 1 (*npr-1*) mutants. The laboratory reference strain N2, on the other hand, has a gain-of-function mutation in the *npr-1* gene that suppresses aggregation (*de Bono and Bargmann, 1998*), rendering N2 animals solitary feeders. Thus, a small genetic difference (just two base pairs in one gene for the *npr-1(ad609lf)* mutant) has a big effect on the population-level behavioral phenotype. Previous research on collective feeding has focused primarily on the genetics and neural circuits that govern aggregation (*Bretscher et al., 2008*; *Busch et al., 2012*; *Chang et al., 2006*; *Cheung et al., 2005*; *de Bono et al., 2002*; *de Bono and Bargmann, 1998*; *Gray et al., 2004*; *Jang et al., 2017*; *Macosko et al., 2009*), rather than on a detailed understanding of the behavior itself. *Rogers et al. (2006)* is a notable exception and includes an investigation of the behavioral motifs that might lead to cluster formation including direction reversals at the edge of clusters. However, we do not know whether these candidate motifs are sufficient to produce aggregation. We also do not know whether aggregation at short times and swarming at longer times are distinct behaviors or different emergent properties of the same underlying phenomenon.

In this paper, we use fluorescence imaging and multi-worm tracking to examine individual behavior inside aggregates. We present new and systematic quantification of the aggregation behavior in hyper-social *npr-1(ad609lf)* mutants (henceforth referred to as *npr-1* mutants) and hypo-social N2 worms. Next, we draw on the concept of motility-induced phase transitions to explain aggregation as an emergent phenomenon by modulating only a few biophysical parameters. Unlike aggregation driven by attractive forces, in motility-induced phase transitions individuals can also aggregate simply due to their active movement and non-attractive interactions, such as volume exclusion (avoidance of direct overlap) (*Redner et al., 2013a*). For instance, this concept has contributed understanding to the aggregation of rod-shaped *Myxococcus xanthus* bacteria, which, similar to *C. elegans*, also exhibit reversals during aggregation (*Mercier and Mignot, 2016*; *Peruani et al., 2012*; *Starruß et al., 2012*). We build an agent-based phenomenological model of simplified worm motility and interactions. By mapping out a phase diagram of behavioral phenotypes, we show that

modulating cluster-edge reversals and a density-dependent switch between crawling speeds are sufficient to produce some aggregation, but not the compact clusters observed in experiments. We found that medium-range taxis towards neighboring worms is necessary to tighten clusters and increase persistence. Finally, combining this model with food depletion gives rise to swarming over time, suggesting that the same behavioral rules that lead to the initial formation of aggregates also underlie the dynamic swarming reported here.

## Results

### Dynamic swarming occurs in social worms at long time scales

Aggregation has most often been characterized as the fraction of worms inside clusters, where individual worms can move in and out of clusters. Here we report an additional dynamic swarming phenotype in aggregating *C. elegans* that occurs on a timescale of hours. Here, swarming refers to the collective movement of a coherent group of worms across a bacterial lawn (*Figure 1A*, *Video 1*). Because of the long timescale, this behavior is not obvious from manual observations of worms on a plate, but becomes clear in time lapse videos (*Figure 1B and C*, *npr-1* panels). Even though N2 worms do not swarm in our experiments (*Figure 1B and C*, N2 panels), they can swarm under appropriate conditions, such as when a clonal population has depleted almost all food (*Hodgkin and Barnes, 1991*) or on unpalatable *Pseudomonas fluorescens* bacterial lawns (personal communication from J. Hodgkin and G.M. Preston). Thus swarming in *C. elegans* does not require loss of *npr-1* function in all environments.

Dynamic swarming occurs with just 40 *npr-1* mutants (*Figure 1B*, top row), making it experimentally feasible to study. Usually a single *npr-1* aggregate forms on the food patch and then moves around the lawn in a persistent but not necessarily directed manner (*Figure 1C*, left; *Figure 1—figure supplement 1*), at a steady speed (*Figure 1D*). The onset of this collective movement appears to coincide with local food depletion, and continues until complete food depletion, at which time the cluster disperses. More than one moving cluster may co-exist, and occasionally a cluster may disperse and form elsewhere when it crosses its previous path (*Figure 1—figure supplement 1*), presumably due to local food depletion. The observed pattern of *npr-1* cluster motion is reminiscent of a self-avoiding, persistent random walk (i.e. not returning to areas that the worms have previously been where there is no food left). By contrast, after initially forming transient clusters on the lawn, N2 worms move radially outwards with no collective movement (*Figure 1C*, right).

### Fluorescence imaging and automated animal tracking allows quantification of dynamics inside and outside of aggregates

Based on our observation that swarming appears to be driven by food depletion, we hypothesize the phenomenon may be a dynamic extension of the initial aggregation that occurs before depletion. To test this idea, we first sought to identify the mechanisms underlying aggregation.

The presence of aggregates is clear in bright field images, but it is difficult to track individual animals in these strongly overlapping groups for quantitative behavioral analysis. We therefore labeled the pharynx of worms with green fluorescent protein (GFP) and used fluorescence imaging in order to minimize overlap between animals (*Video 2*), making it possible to track most individuals even when they are inside a dense cluster (*Figure 2A*). We also labeled a small number of worms (1–3 animals out of 40 per experiment) with a red fluorescent protein (RFP)-tagged body wall muscle marker instead of pharynx-GFP. These RFP-labeled worms were recorded on a separate channel during simultaneous two-color imaging (*Figure 2B*), thus allowing both longer trajectories and the full posture to be obtained in a subset of animals. We wrote a custom module for Tierpsy Tracker (*Javer et al., 2018*) to segment light objects on a dark background and to identify the anterior end of the marked animals automatically, in order to extract trajectories and skeletons of multiple worms from our data (*Figure 2C*).

### Ascarosides and direct adhesion are unlikely to drive different aggregation phenotypes

We first considered long-range chemotaxis driven by food or diffusible ascaroside pheromone signals as a potential behavioral mechanism. Chemotaxis towards food can likely be ignored as our

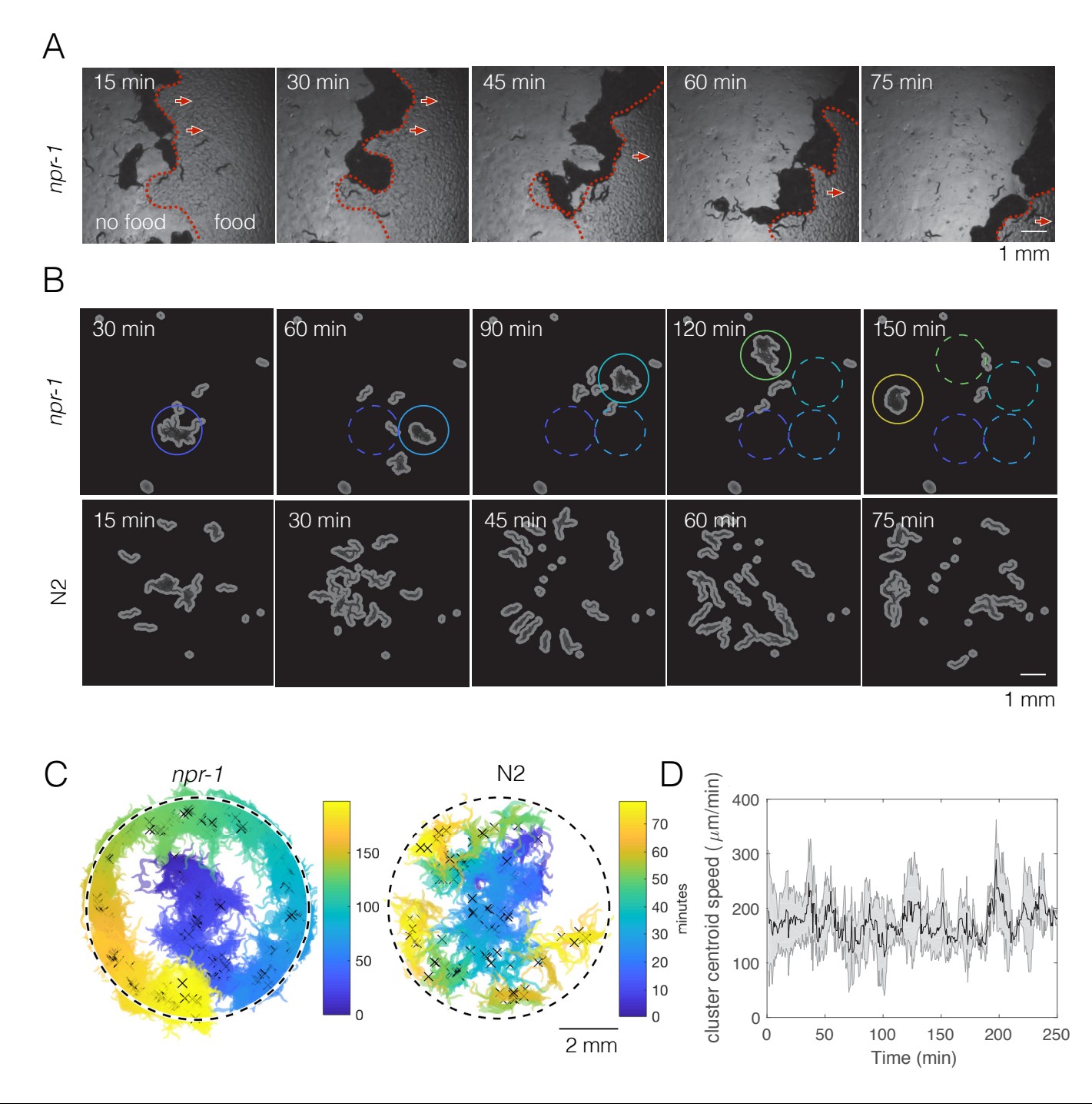

**Figure 1.** *npr-1* but not N2 worms show swarming behavior over time on thin bacterial lawn. (**A**) A few hundred *npr-1* mutant worms form dense clusters that move on food over time. Red dashed lines show the food boundary, where area with food is to the right and food-depleted area is to the left; red arrows show the direction of cluster movement. (**B**) Forty *npr-1* mutant worms also cluster and swarm on food. Solid circles encompass the same cluster at different time points; dashed circles show cluster positions prior to the current time point. The same number of N2 worms do not swarm under our experimental conditions, and instead disperse after initial transient aggregation. (**C**) Visualization of persistent swarming over time. One frame was sampled every 30 s over the duration of the videos and binary segmentation was applied using an intensity threshold to separate worm pixels from the background. Blobs with areas above a threshold value were plotted as clusters to show cluster position over time. The same videos as in (**B**) were used. Dashed circles show the food boundary. Crosses are cluster centroids at each sample frame. (**D**) Centroid speed of persistent *npr-1* clusters, calculated from centroid positions as indicated in (**C**) and smoothed over 10 min. Shaded area shows standard deviation across five replicates.

*Figure 1 continued on next page*

*Figure 1 continued*

DOI: https://doi.org/10.7554/eLife.43318.003

The following figure supplements are available for figure 1:

**Figure supplement 1.** *npr-1* swarming on a bigger food patch.

DOI: https://doi.org/10.7554/eLife.43318.004

**Figure supplement 2.** Stereotyped temporal dynamics.

DOI: https://doi.org/10.7554/eLife.43318.005

experiments were performed on thin, even bacterial lawns, and worms are mostly on food during the aggregation phase of the experiments (99.7 ± 0.4% for *npr-1* and 99.8 ± 0.3% for N2, mean ±S. D.). Although ascarosides are important for processes such as mating and dauer formation in *C. elegans* (*Srinivasan et al., 2008*), it is less clear whether long-range signaling via pheromones plays a role in aggregation (*de Bono et al., 2002*; *Macosko et al., 2009*). *daf-22(m130)* mutants do not produce ascarosides, but *daf-22;npr-1* double mutants aggregate similarly to *npr-1* single mutants (*Figure 3—figure supplement 1*), consistent with the observation that the hermaphrodite-attractive pheromone icas#3 is attractive to both N2 animals and *npr-1* mutants (*Srinivasan et al., 2012*) and is thus unlikely to explain the difference in their propensity to aggregate. Moreover, attraction between moving objects is known to produce aggregation in active matter systems (*Redner et al., 2013a*), but it is not known whether this applies to worms. Short-range attraction between worms may exist in the form of adhesion mediated through a liquid film (*Gart et al., 2011*), but we have no reason to believe this would differ between *npr-1* and N2 strains.

## Reversal rates and speed depend on neighbor density more strongly in *npr-1* mutants than in N2

Having considered long-range food- or ascaroside-mediated attraction and short-range adhesion, we next focused on behavioral responses to nearby neighbors. While postural changes do not seem to be a main driver of aggregation as principal component analysis of lone versus in-cluster *npr-1* worms revealed similar amplitudes in the posture modes (*Figure 3—figure supplement 2*), we found experimental evidence for density-dependence of both reversal rates and speed and that these differ between the two strains we studied.

Reversals have been previously suggested as a behavior that may enable *npr-1* worms to stay in aggregates (*Rogers et al., 2006*). To avoid cluster definitions based on thresholding the distance between worms, we quantified individual worm behavior as a function of local density (*Figure 3A*) instead. Calculating the reversal rates relative to that of worms at low densities, we found that *npr-1* mutants reverse more at increased neighbor densities, while N2 animals do not (*Figure 3B*).

Next we calculated the speed distributions of individual worms, binned by local neighbor density. We found that both strains slow down when surrounded by many other worms, but the shift is more pronounced for *npr-1* animals. *npr-1* worms move faster than N2 at low densities, showing a distinct peak at high speeds. As neighbor density increases, this high speed peak gradually becomes replaced by a peak at low speeds, so that the overall speed distribution for *npr-1* resembles that of N2 at very high densities. Thus, *npr-1* and N2 animals show different density-dependent changes in their respective speed profiles (*Figure 3C*).

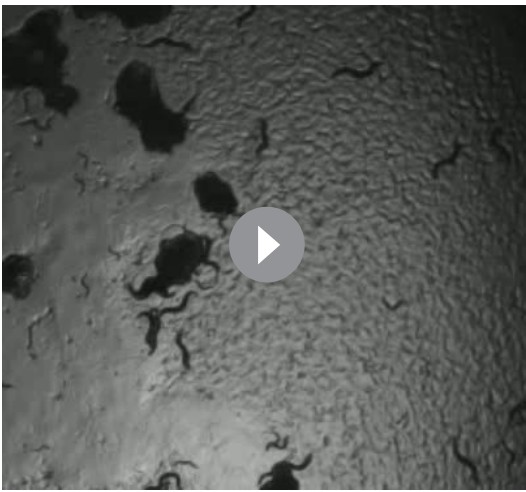

**Video 1.** Sample video showing *npr-1* collective feeding dynamics (bright field high-number swarming imaging). The video plays at 300x the normal speed.

DOI: https://doi.org/10.7554/eLife.43318.006

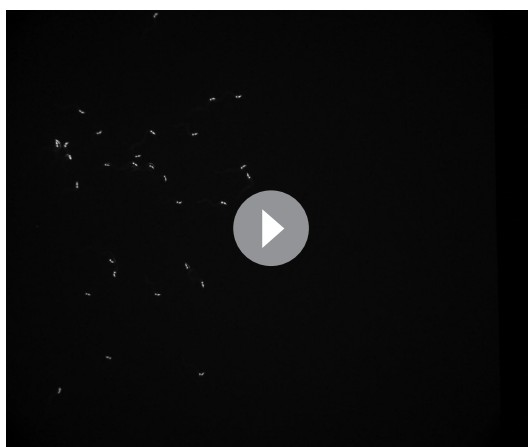

**Video 2.** Sample video showing *npr-1* collective feeding dynamics (fluorescence 40 worm aggregation imaging). The video plays at 90x the normal speed.
DOI: https://doi.org/10.7554/eLife.43318.008

Since the observed transition of the speed profiles could occur due to active behavioral changes as well as restricted movement in clusters, we also considered tracks of individual worms. Using body wall muscle-marked worms allowed us to obtain longer trajectories that could be joined for the duration of an entire video, including cluster entry and exit events. We compared the speed of these tracks with visual assessment of when a worm entered or exited a cluster based on the proximity to pharynx-labeled worms. We found that worms are able to move inside of clusters and observed that speed changes can occur prior to cluster entry and exit events (*Figure 3D*, *Video 3* and *Video 4*). This change of speed is neither purely mechanical nor a deterministic response to a certain neighbor density, and suggests a mechanism in which worms probabilistically switch between different speeds.

## Spatial statistics show group-level differences between *npr-1* and N2 animals

The differences in aggregation behavior between *npr*-1 and N2 are visually striking, but previous quantification has typically been limited to the fraction of animals in clusters. Using the tracked positions of pharynx-labeled worms (*Figure 4A*), we calculated the pair-correlation function (*Figure 4B*), commonly used to quantify aggregation in cellular and physical systems (*Gurry et al., 2009*). We

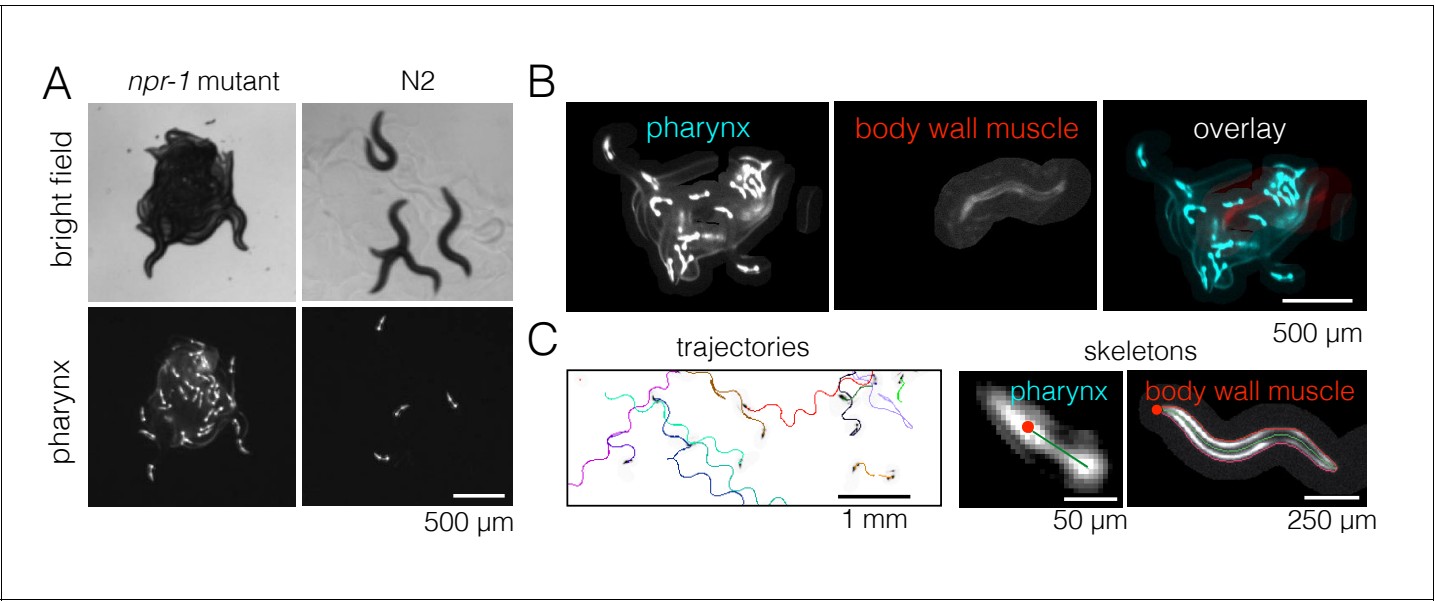

**Figure 2.** Fluorescence multi-worm tracking. (**A**) *npr-1* mutant and N2 animals exhibit different social behaviors on food, with the former being hyper-social (top left) and the latter being hypo-social (top right). Using a pharynx-GFP label (bottom row), individual animals may be followed inside a cluster. (**B**) In two-color experiments, worms are either labeled with pharynx-GFP (left) or body wall muscle-RFP (middle). As the two colors are simultaneously acquired on separate channels, the selected few RFP-labeled individuals are readily segmented and may be tracked for a long time, even inside a dense cluster. (**C**) Tierpsy Tracker tracks multiple worms simultaneously, generating both centroid trajectories (left, image color inverted for easier visualization; multiple colors show distinct trajectories) and skeletons (middle, pharynx-marked animal; right, body wall muscle-marked animal; red dots denote the head nodes of the skeleton).
DOI: https://doi.org/10.7554/eLife.43318.007

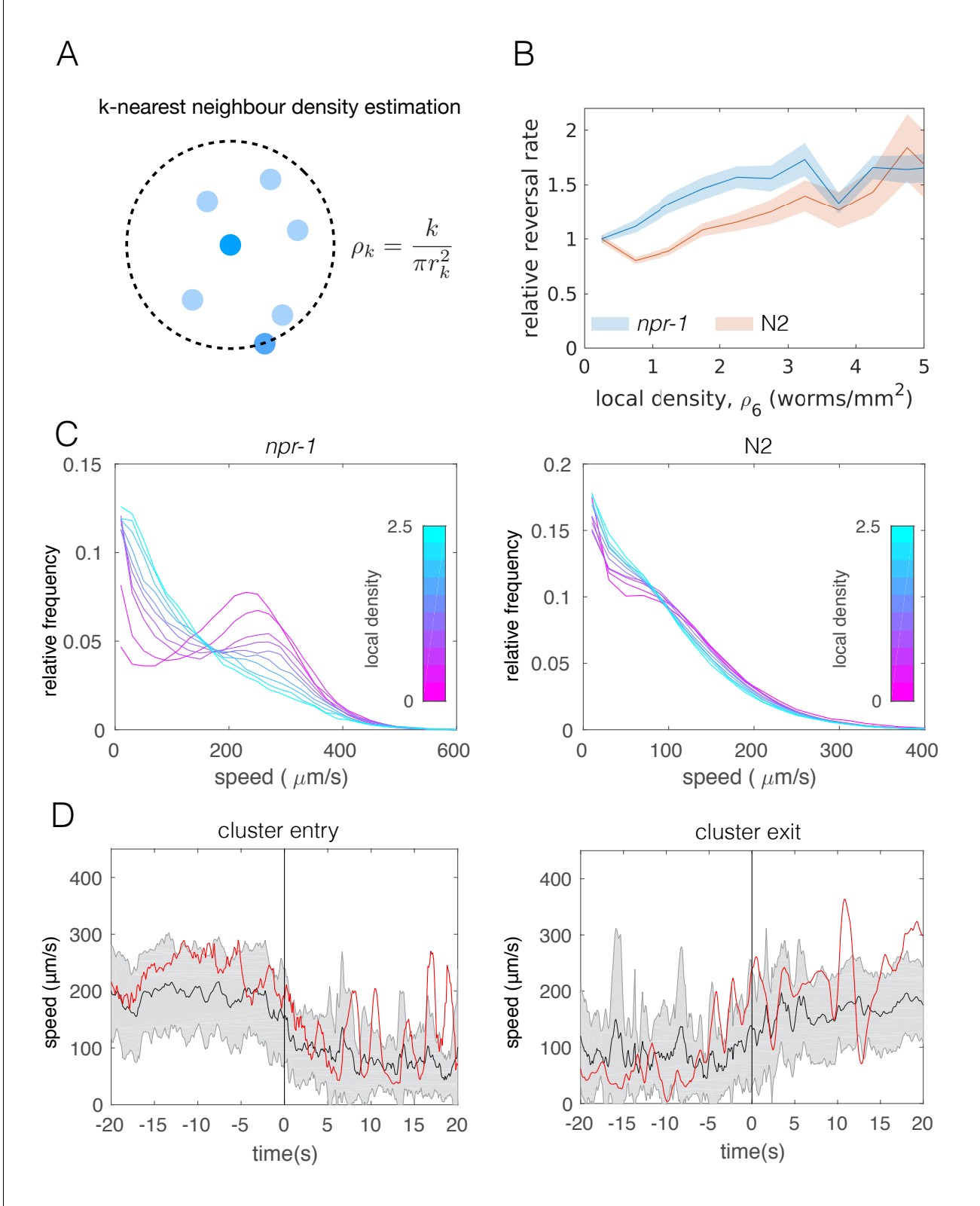

**Figure 3.** Individual-level behavioral quantification. (**A**) Schematic explaining *k*-nearest neighbor density estimation. (**B**) Relative rate of reversals as a function of local density (*k*-nearest neighbor density estimation with *k* = 6) for *npr-1* (blue) and N2 (orange) strains. Lines show means and shaded area shows the standard error (bootstrap estimate, 100 samples with replacement). (**C**) Distributions of crawling speeds at different local neighbor densities for both strains. Lines show histograms of speeds for each density bin, and the color of the line indicates the density (blue is high, magenta is low). (**D**)
*Figure 3 continued on next page*

*Figure 3 continued*

Midbody absolute speed for manually annotated *npr-1* cluster entry (left, n = 28) and exit events (right, n = 29). Each event was manually identified, with time 0 representing the point where the head or tail of a worm starts to enter (left) or exit (right) an existing cluster. Skeleton *xy*-coordinates were linearly interpolated for missing frames for each event, before being used to calculate midbody speed extending 20 s on both sides of time 0 of the event. Speeds were smoothed over a one-second window. Shading represents standard deviation across events. Each red line shows the midbody absolute speed of a selected event that is shown in *Video 3* (left) or *Video 4* (right).

DOI: https://doi.org/10.7554/eLife.43318.009

The following figure supplements are available for figure 3:

**Figure supplement 1.** Pheromones appear unimportant for aggregation.
DOI: https://doi.org/10.7554/eLife.43318.010
**Figure supplement 2.** Shape analysis for lone and in-cluster *npr-1* worms.
DOI: https://doi.org/10.7554/eLife.43318.011

also computed a hierarchical clustering of worm positions (*Figure 4C*), which is calculated from the same pairwise distances but emphasizes larger scale structure. Using both measures, we found that as a population, *npr-1* animals show quantifiably higher levels of aggregation than N2, especially at scales up to 1 mm (pair-correlation '$S_1$', *Figure 4D*) and 2 mm (hierarchical clustering '$S_2$', *Figure 4E*). We also quantified aggregation using scalar spatial statistics, namely the average standard deviation ('$S_3$') and kurtosis ('$S_4$') of the distribution of positions. This confirms that the positions of *npr-1* worms are less spread-out and more heavy-tailed than those of N2 (*Figure 4D*).

## Agent-based model captures different aggregation phenotypes

To test whether the individual behavioral differences measured between *npr-1* and N2 worms are sufficient to give rise to the observed differences in aggregation, we constructed a phenomenological model of worm movement and interactions. The model is made up of self-propelled agents (*Figure 5A*), and includes density-dependent interactions motivated by the experimental data, namely reversals at the edge of a cluster (*Figure 5B*) and a switch between movement at different speeds (*Figure 5C*). As a model of collective behavior this differs from those commonly considered in the literature, such as the Vicsek model (*Vicsek et al., 1995*) and its many related variants (*Vicsek and Zafeiris, 2012*; *Yates et al., 2011*). Such models typically feature attractive forces or align the direction of motion at ranges much longer than the size of the moving objects, and result in flocking or clustering with global alignment (*Figure 5D*), which we do not observe in our experimental data. In contrast, our model needs to produce dynamic, disordered aggregates (*Figure 1B*, *Figure 2A* and *Video 2*), and should primarily rely on short-range interactions that are motivated by behaviors measured in our data.

The density-dependence of the reversal rate and speed switching is implemented as follows: The rate of reversals increases linearly with density with slope $r'$, which is a free parameter, and is thus given by $r_{rev} = r' \rho$. The reversal rate at zero density is zero as we ignored spontaneous reversals outside of clusters as these were only rarely observed under our experimental conditions (see Appendix 1 for further discussion of the model construction). This parameterization of the reversal rate may be unbounded, but we can prevent unrealistically high reversal rates for a given maximum worm number by choosing our prior distribution of the parameter $r'$. The rate of slowing down is similarly approximated as a linear function of density, with free parameter $k_s'$, and is given by $k_{slow} = k_{s0} + k_s' \rho$, where $k_{s0}$ is the slowing rate at zero-density. The rate of speeding up is given by $k_{fast} = k_{f0} \exp[-k_f' \rho]$, where the exponential decay is chosen to ensure positivity

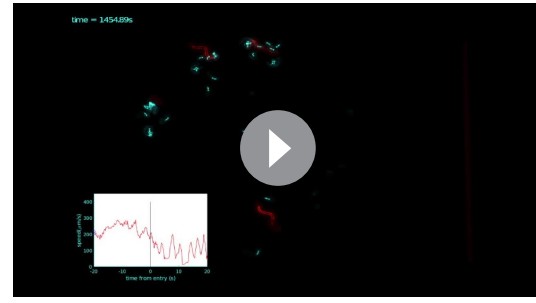

**Video 3.** A single event showing switch from high to low motility state prior to cluster entry (fluorescence 40 worm aggregation imaging). The red worm at the bottom (arrow) decreases speed before entering a cluster. Inset: midbody absolute speed of that individual with respect to time 0 as the point of the head entering a cluster; open blue circle shows the current speed matched to the video frame.
DOI: https://doi.org/10.7554/eLife.43318.012

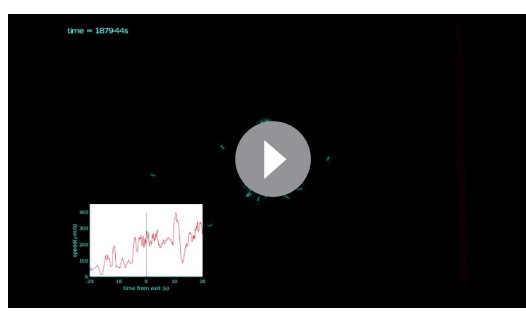

**Video 4.** A single event showing switch from low to high motility state prior to cluster exit (fluorescence 40 worm aggregation imaging). The red worm increases speed before exiting a cluster. Inset: midbody absolute speed of that individual with respect to time 0 as the point of the head exiting a cluster; open blue circle shows the current speed matched to the video frame.
DOI: https://doi.org/10.7554/eLife.43318.013

of the rate, and $k_{f0}$ is the rate at zero density. The rates of slowing down and speeding up at zero density ($k_{s0}$, $k_{f0}$) were obtained from published single-worm experimental data (*Javer et al., 2018*; *Yemini et al., 2013*).

We initially ran a coarse parameter sweep, sampling uniformly in the two-dimensional parameter space associated with the density-dependence of reversals and speed switching. As a simplifying assumption, the density-dependence of the speeding-up and slowing-down rates was set equal ($k'_s = k'_f = k'$). The remaining parameters, $r'$ and $k'$, were varied to explore the global model behavior. This demonstrates that our model can capture different aggregation phenotypes from solitary movement to aggregation (*Figure 5E*) by varying just two free parameters, and provides important general insights. Inspection of the model simulations shows that each behavior alone (just reversals or slowing) does not give the same level of aggregation as when both parameters are modulated (*Figure 5E*), so that using both behavioral components proves important. Quantifying the aggregation and comparing it to the *npr-1* experiment, however, highlights incomplete quantitative agreement with both the pair correlation function and hierarchical clustering distribution (*Figure 5F*). Thus, we reasoned additional interactions may be required to match the experimentally observed behaviors.

## Adding a medium-range taxis interaction promotes stronger aggregation

To explore improvements in clustering, we extended the model by an attractive taxis interaction. Attraction should intuitively improve clustering, but we knew from our model exploration that an attractive potential between bodies produces undesirable cluster shapes (*Figure 5D*) and reasoned that a long-range interaction may be unrealistic (*Figure 3—figure supplement 1*). Thus, we include taxis towards neighboring worms and model worm movement as an attractive persistent random walk. The taxis contribution to a worm's motile force has an overall strength controlled by parameter $f_t$, with multiple nearby neighbors contributing cumulatively, weighted by $1/r$, where $r$ is the distance to a neighboring worm. Neighboring worms beyond a cut-off distance equal to the length of a worm have no contribution. Thus, this taxis interaction is acting at a natural intermediate length scale of our system (see Appendix 1 for details).

The resulting extended model has four free parameters: density-dependent reversals ($r'$), speed-switching rates ($k'_s$, $k'_f$) and taxis ($f_t$). To find the parameter combinations that best describe each strain, as well as the uncertainty in the parameter values, we used an approximate Bayesian inference approach (see Appendix 1). To increase the computational efficiency of our inference pipeline, we excluded infeasible regions of parameter space to reduce the prior distribution of parameters that we need to sample from (*Figure 6—figure supplement 1*) (see Appendix 1). We then selected the closest matching simulations from about 27,000 simulations for *npr*-1 and about 13,000 simulations for N2, equally weighting all four summary statistics. Results from our extended model (*Figure 6A*, *Video 5* and *Video 6*) show markedly improved quantitative agreement with the experiments (*Figure 6B*). The approximated posterior distributions of the parameters (*Figure 6C–D*) show the most likely values of the parameters for each strain, as well as the uncertainty associated with the individual and joint marginal parameter distributions. In particular, to achieve *npr-1*-like aggregation, the reversal ($r'$) and taxis ($f_t$) parameters need to be higher than for N2, albeit not too high. The density-dependence of the slowing rate ($k'_s$) is only subtly different between the two strains, while the dependence of the speeding up rate ($k'_f$) is greater in *npr-1*, but with broader uncertainty.

To address whether all three behaviors (reversals, speed changes, and taxis) were necessary for aggregation we ran additional simulations: starting from the mean of the posterior distribution for *npr-1* (*Figure 6C*) as a reference, we removed individual model components by setting the

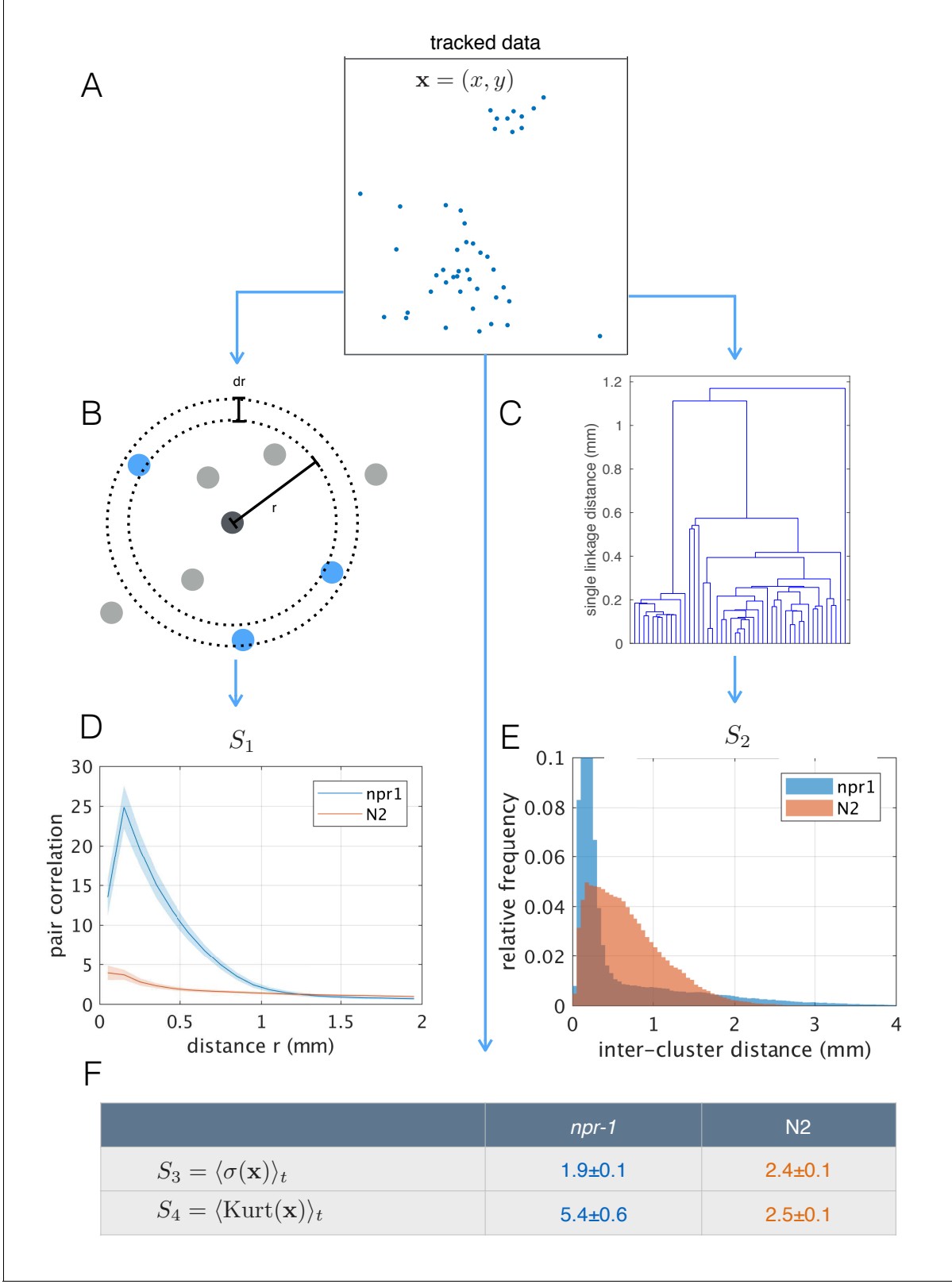

**Figure 4.** Population-level behavioral quantification. (A) Positions of *npr-1* worms in an example frame. (B) Schematic explaining pair correlation function ($S_1$), which counts the number of neighbors at a distance $r$, normalized by the expectation for a uniform distribution. (C) Example dendrogram from which hierarchical clustering branch length distributions ($S_2$) can be calculated. (D) Pair correlation function for *npr-1* (blue) and N2 (orange). Lines show mean and shaded area shows standard error of the mean. (E) Hierarchical clustering branch length distributions for *npr-1* (blue) and N2 (orange). *Figure 4 continued on next page*

*Figure 4 continued*

Histograms show relative frequency of inter-cluster distances (single linkage distance in agglomerative hierarchical clustering, equivalent to the branch lengths in the example dendrogram in (C)). (**F**) Mean standard deviation ($S_3$) and kurtosis ($S_4$) of the positions of worms, with the mean taken over frames sampled.

DOI: https://doi.org/10.7554/eLife.43318.014

corresponding parameters to zero. These perturbed simulations show that removing speed switching or taxis from the model disrupts aggregation, while removing reversals reduces the overall quantitative agreement with experimental data (*Figure 6—figure supplement 2, B–D*). In some cases, removing individual model behaviors also produced correlations of velocity and orientation between neighbors that are different from what we measure in experiments (*Figure 6—figure supplement 3*). Thus, we conclude that we have identified sufficient behavioral components for aggregation, and that these are also necessary to quantitatively match aggregation in *npr-1* mutants.

Searching for evidence of taxis in the experimental tracking data, we calculated the correlation between worm velocity and the vector towards nearby worms, and found this correlation to be nearly zero in both experiments and simulations for all distances up to 2 mm (*Figure 6—figure supplement 3B1–2*), which is larger than the size of a typical worm cluster. This may not be intuitive, and we suspect the reason is twofold: (a) the taxis effect is only a small influence on the instantaneous direction of the movement of a worm, compared to persistence and noise; and (b) we only tracked the pharynx in our experiments, and reproduced this restriction in our analysis of simulations, but the whole body of the worm is likely giving relevant cues to any chemical or mechanical taxis. Our methodology that enables us to track inside worm clusters therefore brings with it the caveat that there is unseen worm density that affects any potential taxis behavior, but which remains undetectable in our tracking. Thus, our analysis shows that a taxis behavior similar to our simulations may be present in experiments, even if it is difficult to detect with correlation analysis. We compared the other inferred parameters with experimental measurements: The reversal rate shows a similar increase with density that is greater for *npr-1* than N2 (*Figure 6—figure supplement 4B*). The speed switching rates could only be compared indirectly by calculating the ratio of fraction of worms in fast vs. slow movement in experiments (*Figure 6—figure supplement 4C1*) and model simulations (*Figure 6—figure supplement 4C2*). The disagreement may indicate that the exponential form of $k_f(\rho)$ is only a rough approximation. However, aggregation in the model is not sensitive to speed switching rates, as shown by the broad posterior distributions for the inferred parameters (*Figure 6C–D*).

## Extending the model with food-depletion captures dynamic swarming

Since we hypothesize that the swarming we observed at longer time scales may be explained as aggregation under food depletion conditions, we further extended the model to allow the local depletion of food. Food is initially distributed uniformly, and becomes depleted locally by worm feeding (see Appendix 1 for details). Absence of food suppresses the switch to slow speeds, thus causing worms to speed up when food is locally depleted. As a result, we hypothesize that worm clusters begin to disperse but reform on nearby food, leading to sweeping.

Selecting the parameter combination best matching the *npr*-1 strain (*Figure 6*) and an appropriate food depletion rate (chosen such that all food was depleted no faster than observed in experiments), the resulting simulation produced long-time dynamics qualitatively representative of the experimentally observed swarming (*Figure 7A–B*, *Video 7*). Worm clusters undergo a persistent but not necessarily directed random walk, can disperse and re-form elsewhere, and multiple clusters may co-exist, all of which we observe experimentally. Tracking the centroid of worms in our simulations, we find a comparable cluster speed as the median experimental value of 172 µm/min (*Figure 1D*) for a range of feeding rates (*Figure 7C*) (feeding rate is an unknown parameter as our model only accounts for relative food concentration). Thus, the model indicates that dynamic swarming of *npr*-1 aggregates may be explained as an emergent phenomenon resulting from individual locomotion, and that the same behavioral mechanisms that produce the initial aggregates, when coupled with local food depletion, give rise to the observed swarming behavior.

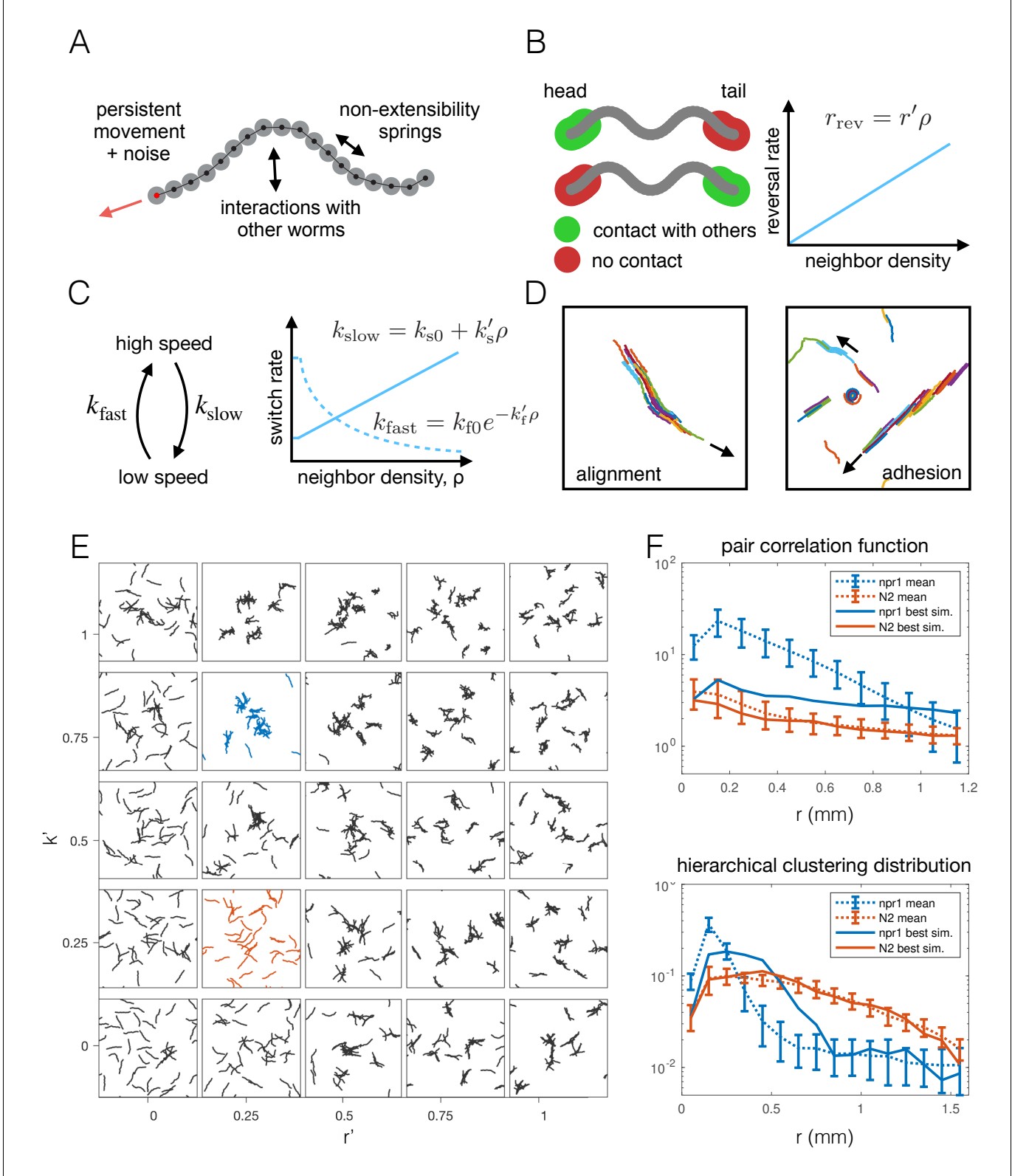

**Figure 5.** Agent-based modeling of emergent behavior. (A) Schematic of individual worm in the agent-based model. Each worm is made up of $M$ nodes (here $M = 18$), connected by springs to enforce non-extensibility. Each node undergoes self-propelled movement, with the head node (red dot) undergoing a persistent random walk, and the rest of the nodes follow in the direction of the body. (B) Schematic of simulated reversals upon exiting a cluster. Each worm registers contact at the first and last 10% of its nodes within a short interaction radius. If contact is registered at one end but not the

*Figure 5 continued on next page*

*Figure 5 continued*

other, the worm is leaving a cluster and thus reverses with a Poisson rate dependent on the local density. (C) Schematic of density-dependent switching between movement speeds. Worms stochastically switch between slow and fast movement with Poisson rates $k_{slow}$ and $k_{fast}$, which increase linearly and decrease exponentially with neighbor density, respectively. (D) Snapshots of simulations with commonly considered aggregation mechanisms, which produce unrealistic behavior for worm simulations, with flocking and highly aligned clustering. Arrows indicate the direction of movement of large clusters. (E) Phase portrait of model simulations, showing snapshots from the last 10% of each simulation, for different values of the two free parameters: density-dependence of the reversal rate and density-dependence of speed-switching (here $k_{slow} = k_{fast}$). Blue and orange panels highlight best fit for *npr-1* and N2 data, respectively. (F) Summary statistics $S_1$ (pair correlation, top) and $S_2$ (hierarchical clustering, bottom) for the simulation which most closely matches the experimental data for the *npr-1* and N2 strains (blue and orange panels in (E), respectively).
DOI: https://doi.org/10.7554/eLife.43318.015

## Discussion

We have investigated the mechanisms of aggregation and swarming in *C. elegans* collective feeding using quantitative imaging and computational modeling. We show that while a combination of increased reversals upon leaving aggregates and a neighbor density-dependent increase in speed switching rates is sufficient to produce aggregation, the addition of taxis towards neighbors improves the quantitative agreement between simulations and experiments. Removing any one of the core behavioral mechanisms (reversals, speed changes, taxis) from our model either disrupts aggregation or otherwise reduces the quantitative agreement with experiments (*Figure 6—figure supplement 2–3*). The proposed taxis might be driven by a shallow $O_2$ or $CO_2$ gradient created by a worm cluster (discussed further below), to additional chemical signals unaffected by *daf-22* loss of function, or to another unknown mechanism. By extending the aggregation model to include food depletion, we show that the same behavioral mechanisms also underlie dynamic swarming in the hyper-social *C. elegans* strain, reminiscent of wild fires and other self-avoiding dynamics.

We focused on identifying phenomenological behavioral components giving rise to aggregation, while remaining agnostic as to the sensory cues causing the behaviors. The density-dependent interactions could arise from local molecular signaling, or be mediated through contact-sensing, and the $1/r$ dependence of the taxis interaction is compatible with a diffusible, non-degrading factor (such as $CO_2$, or $O_2$ depletion; dependence would likely be different for a pheromone depending on its degradation rate). Given that aggregates break up when ambient $O_2$ concentration is reduced to 7% (*Gray et al., 2004*), the preferred concentration of *npr-1* mutants, the most obvious candidate for the sensory cue guiding aggregation is $O_2$ (*Rogers et al., 2006*). A simple hypothesis would be that oxygen consumption by worms locally lowers $O_2$ concentration to the 5–12% preferred by *npr-1* mutants, promoting their aggregation. To support this, *Rogers et al. (2006)* report low $O_2$ concentrations inside worm clusters. However, non-aggregating N2 worms also prefer $O_2$ concentrations lower than atmospheric (5–15%) (*Gray et al., 2004*). Furthermore, a strong reduction of oxygen concentration inside an aggregate to near 7% is unlikely based on reaction-diffusion calculations: the diffusion of oxygen through worm tissue, or their oxygen consumption, would need to be several orders of magnitude different from estimated values to create $O_2$ gradients as steep as reported by Rogers *et al.* (*Appendix 2—figure 1*). However, as worms have been reported to respond even to small changes in oxygen concentration (*McGrath et al., 2009*), aggregation may still be mediated through a shallower local oxygen gradient.

In this scenario, high ambient $O_2$ concentration serves as a permissive signal for aggregation and a shallow oxygen gradient induces worms to stay inside aggregates. Our agent-based simulations are entirely compatible with this picture. Further experiments would be required to test the hypothesis that oxygen is playing such a dual role. One possibility would be to introduce mutations leading to aerobic metabolism deficiencies into *npr-1* mutants. Such mutants would still be able to sense ambient oxygen, but are expected to produce an even weaker oxygen gradient in an aggregate. The resulting phenotype could then be compared quantitatively to model predictions, for example with reduced taxis and/or modified rates of density-dependent reversal and speed switching. Additionally, one may seek evidence for the ability of worms to sense a shallow oxygen gradient by repeating the gas-phase aerotaxis experiment described in *Gray et al. (2004)*, but with a much smaller gradient (19–21%) in the light of our new calculations, to see if worms can sense and move towards environments where oxygen levels are only slightly below ambient concentrations. Further

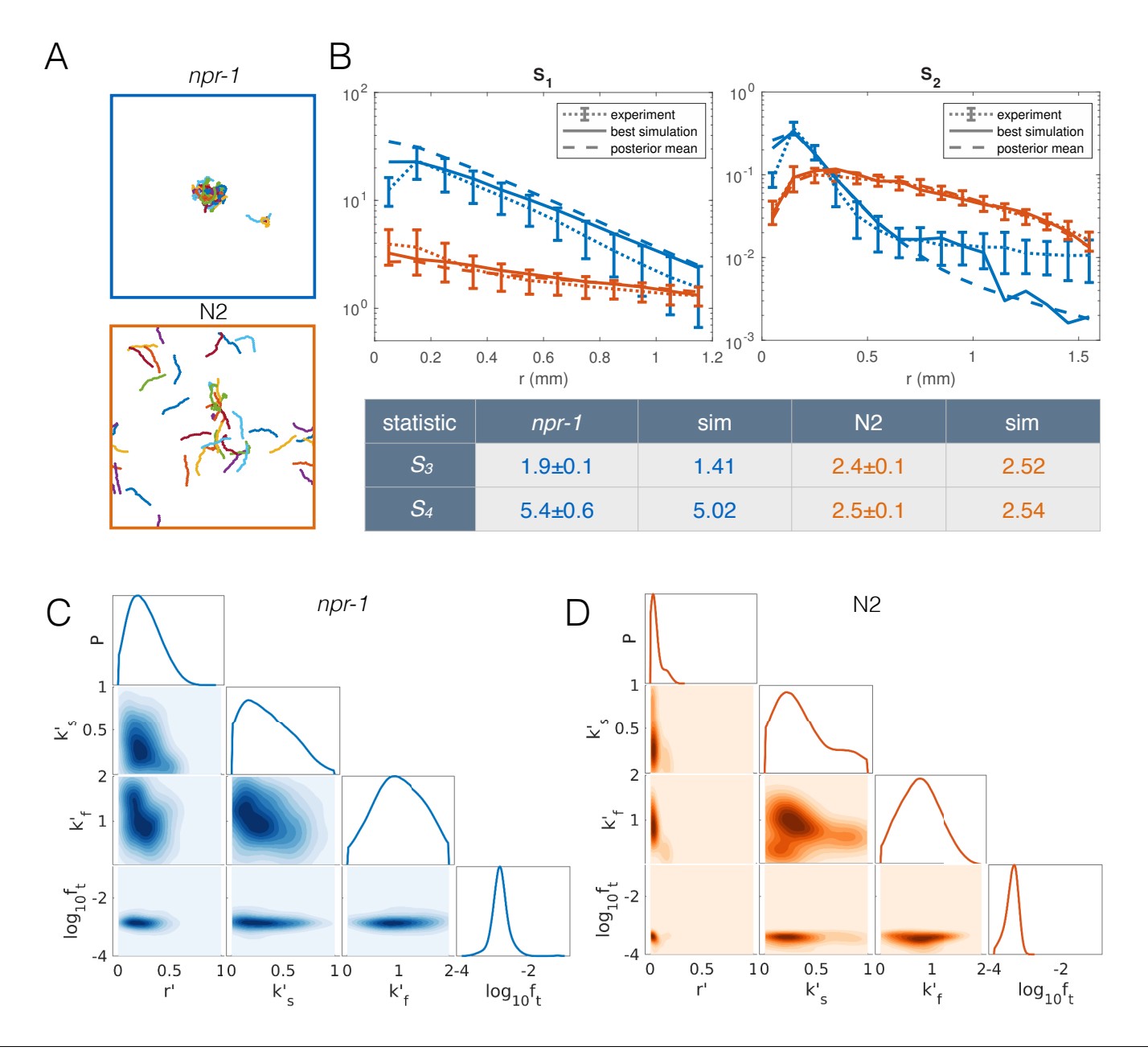

**Figure 6.** Model with taxis captures quantitative aggregation phenotypes. (**A**) Sample snapshot of the closest matching simulations for *npr-1* (top) and N2 (bottom). (**B**) Summary statistics for *npr-1* (orange) and N2 (blue): $S_1$: pair correlation function; $S_2$: hierarchical clustering distribution; $S_3$: standard deviation of positions; $S_4$: kurtosis of positions. Solid lines show the closest matching simulations; dashed lines show sample mean over the posterior distribution; and dotted lines show experimental means, with error bars showing standard deviation of 13 (*npr-1*) and 9 (N2) replicates. (**C–D**) Approximate posterior distribution of parameters for *npr-1* (C) and N2 (D). Diagonal plots show marginal distribution of each parameter, off-diagonals show pairwise joint distributions. Parameters are: increase in reversal rate with density, $r'$; increase in rate to slow down, $k'_s$; decrease in rate to speed up, $k'_f$; and contribution of taxis to motile force, $f_t$.

DOI: https://doi.org/10.7554/eLife.43318.016

The following figure supplements are available for figure 6:

**Figure supplement 1.** Reduced prior distribution used for approximate Bayesian inference of extended model.

DOI: https://doi.org/10.7554/eLife.43318.017

**Figure supplement 2.** Core model components, but not noise and undulations in movement, are necessary for quantitative agreement with aggregation summary statistics.

*Figure 6 continued on next page*

*Figure 6 continued*

DOI: https://doi.org/10.7554/eLife.43318.018

**Figure supplement 3.** Analysis of orientational and velocity correlations in experiments and simulations.

DOI: https://doi.org/10.7554/eLife.43318.019

**Figure supplement 4.** Additional comparison of model parameters with experimental measurements.

DOI: https://doi.org/10.7554/eLife.43318.020

**Figure supplement 5.** Aggregation model requires minimum length of simulated worms, and is robust to introducing volume exclusion.

DOI: https://doi.org/10.7554/eLife.43318.021

work quantifying the behavior of individual worms at different oxygen concentrations, such as during oxygen-shift experiments inside flow chambers where single animals experience acute switches between 21% and 19% oxygen, may also help to distinguish oxygen as a direct cue or part of the 'sensory triggers that can initiate social behavior by activating chemotaxis or mechanotaxis' (*Gray et al., 2004*).

The model of worm movement and interactions presented here was chosen for a balance of simplicity and realism, and is not necessarily unique. Our model comprises a persistent random walk of chain-like worms, which were loosely inspired by work on bacterial systems (*Balagam and Igoshin, 2015*). We have adopted Bayesian parameter inference to capture the uncertainty in our parameter estimates, and to enable flexible extension to additional experimental data or comparison of different models in future work. An alternative approach is to be entirely data-driven in the construction of the model and compute interactions between worms directly based on their tracked positions at every time step, as has been done in collective behavior of *Myxococcus xanthus* (*Cotter et al., 2017*; *Zhang et al., 2018*). This approach may require higher worm numbers and improved tracking, to ensure comparably large statistical sample sizes with bacterial studies. We have used experimental data to inform our modeling framework where appropriate (size, shape, speed of agents, and reversal and speed change rates at zero density), and verified that the aggregation outcome is robust and quantitatively similar to experimental results regardless of the amount of noise in the persistent random walk (*Figure 6—figure supplement 2E–G*), or the presence of undulations in agent movement (*Figure 6—figure supplement 2H*). We have further verified that aggregation still occurs with shorter simulated worms (and fewer nodes per worm), given they are long enough to detect a contact difference between head and tail when exiting a cluster, which is required to initiate reversals (*Figure 6—figure supplement 5A*). Lastly, in the model presented here, we have allowed for overlap between worms to reflect a degree of overlap in clusters when worms can crawl over each other. With volume exclusion our model still produces aggregation, although the clusters are less dense and more extended (*Figure 6—figure supplement 5B*).

One advantage of using *C. elegans* to study animal collective behavior is the opportunity to experimentally control and perturb the system. It should be possible to experimentally modify the key behavioral parameters identified in this paper with mutations or acute stimulus delivery in order to test our model. For example, one can introduce a reversal phenotype with *unc-4* mutations, or alter the speed switching rates with mutations that affect the roaming-dwelling transition. Controlled stimulus delivery has already been used in previous oxygen-shift experiments. The resultant experimental outcomes may then be compared to theoretical predictions. Thus, there are ample opportunities for future studies to further integrate experimental and theoretical

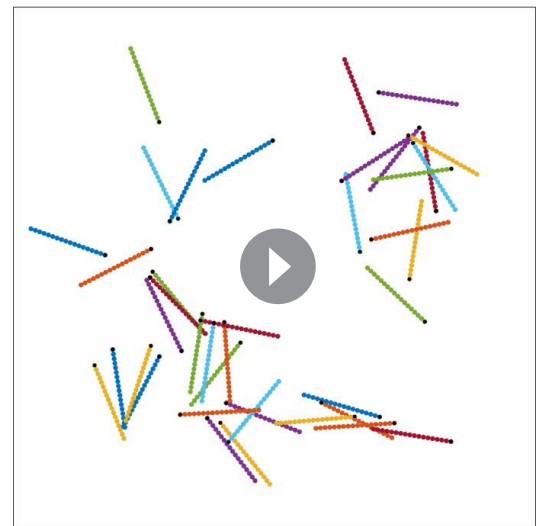

**Video 5.** Sample model (with taxis) simulation describing *npr-1* mutants. The video plays at 30x the normal speed.

DOI: https://doi.org/10.7554/eLife.43318.022

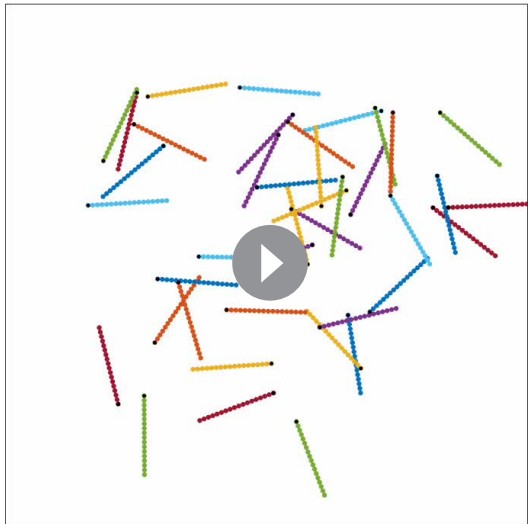

**Video 6.** Sample model (with taxis) simulation describing N2. The video plays at 30x the normal speed.
DOI: https://doi.org/10.7554/eLife.43318.023

methods in the study of *C. elegans* collective behavior.

Despite its extensive study in the lab, it is still uncertain whether aggregation and swarming have a function in the wild. Aggregation may serve to protect *C. elegans* from desiccation or UV radiation associated with the surface environment (*Busch and Olofsson, 2012*). *C. elegans* swarming on unpalatable bacteria may also facilitate predation, perhaps through the collective action of secreted molecules that overcome bacterial defenses (personal communication from J. Hodgkin and G.M. Preston) in a manner similar to the well-described cooperative predation strategy used by *Myxobacteria xanthus* (*Muñoz-Dorado et al., 2016*; *Pérez et al., 2016*). Moreover, social versus solitary foraging strategies may confer selective advantages in different food abundance, food quality, and population density environments (*de Bono and Bargmann, 1998*). The observation that aggregating strains are less fit in laboratory conditions (*Andersen et al., 2014*) suggested that social feeding is not an efficient strategy at least in abundant food conditions. However, the observed fitness difference between aggregating and non-aggregating strains is actually dissociable from the feeding strategy in the lab (*Zhao et al., 2018*), leaving the question unresolved. Furthermore, in other systems, social feeding can increase fitness in natural environments via improved food detection and intake (*Cvikel et al., 2015*; *Li et al., 2014*; *Snijders et al., 2018*). It would be time consuming to experimentally measure the feeding efficiency of different behavioral strategies for a wide range of food patch sizes, distributions, and qualities. The agent-based model used in this study presents an opportunity to use a complementary approach to finding conditions that may favor social feeding.

*C. elegans* bridges the gap between the commonly studied micro- and macro-scales, and finding the behavioral rules underlying this mesoscale system allows us to consider principles governing collective behavior across scales. Indeed, key behavioral rules identified here for *C. elegans* aggregation have been observed at other scales. Spontaneous reversals have been implicated in bacterial aggregation at the microscale (*Mercier and Mignot, 2016*; *Starruß et al., 2012*; *Thutupalli et al., 2015*). By contrast, aggregating worms reverse mainly in response to leaving a cluster rather than spontaneously, thus requiring more complex sensory processing and behavioral response than seen in bacterial systems. Furthermore, changes in movement speed are a common feature in motility-induced phase transitions (*Großmann et al., 2016*; *Redner et al., 2013b*; *Abaurrea Velasco et al., 2018*). The emergent phenomena observed in models of interacting particles generally range from diffusion-limited aggregation to jamming at high volume fractions to flocking of self-propelled rods through volume exclusion (in two-dimensions). In contrast, aggregation in *C. elegans* occurs at much lower numbers of objects (tens of worms) and lower densities (area fraction of 4–6%) than typically studied in this field (thousands of objects at area fractions of 20–80%), and the density dependence of motility changes again emphasizes the role of more complex sensing and behavioral modulations common in macroscale animal groups such as fish shoals (*Ward et al., 2011*). Thus, collective behavior of *C. elegans* at the mesoscale indeed draws from both ends of the size scale and complexity spectrum, linking the physical mechanisms familiar from microscopic cellular and active matter systems with the behavioral repertoire of larger multicellular organisms.

Our approach of decomposing aggregation into component behaviors through modeling may also have applications in quantitative genetics beyond the scope of our current study. While hyper-social *npr-1* mutants and hypo-social N2 worms show phenotypic extremes, wild isolates of *C. elegans* aggregate to different degrees (*de Bono and Bargmann, 1998*). Previous work has shown that even a very small increase in the phenotypic dimensionality (from one to two) can reveal

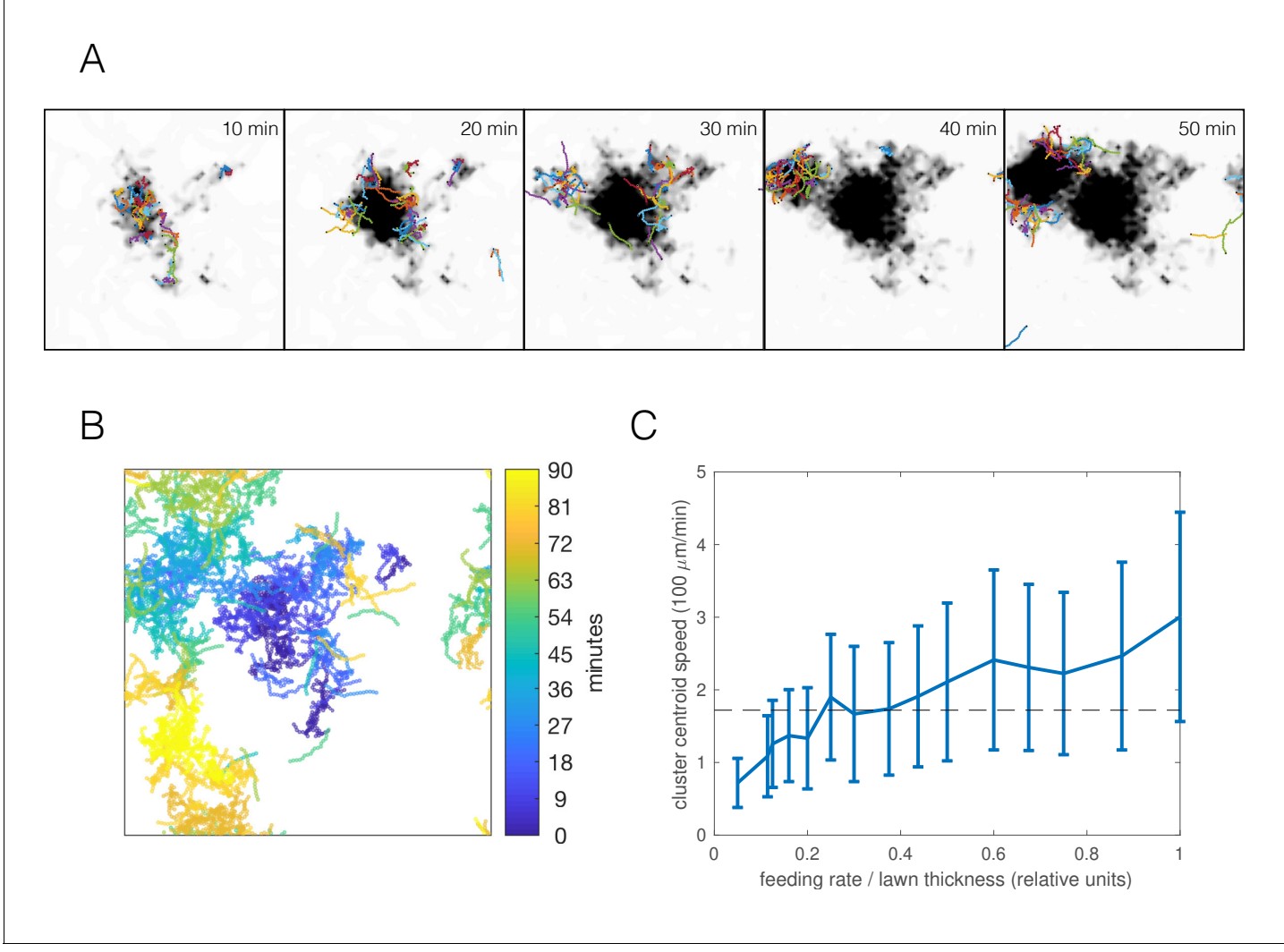

**Figure 7.** Simulations capture dynamic swarming. (**A**) Snapshots of aggregation simulation with food depletion. Background color shows relative food concentration with white indicating high food and black indicating no food. (**B**) Visualization of worm positions in (A) over time, showing cluster displacement. Note the periodic boundary conditions. (**C**) Cluster speed at various feeding rates relative to lawn thickness (other parameters equal to mean of posterior distribution for *npr-1*). The upward trend is expected: smaller lawn thickness leads to faster movement as worms run out of food quicker and need to re-form clusters on nearby food. Cluster speed is calculated the same way as in ***Figure 1D***; error bars show median absolute deviation over five simulations. Dashed line indicates experimentally-derived median cluster speed (from ***Figure 1D***) for comparison.

DOI: https://doi.org/10.7554/eLife.43318.024

independent behavior-modifying loci (***Bendesky et al., 2012***). Thus inferring model parameters for data from multiple wild *C. elegans* strains would produce behavioral parameterizations that might serve as a powerful set of traits for finding further behavior-modifying loci.

## Materials and methods

**Key resources table**

| Resource | Designation | Source or reference | Identifiers | Additional information |
|---|---|---|---|---|
| Strain (*C. elegans*) | N2 | *Caenorhabditis* Genetics Centre | RRID:WB-STRAIN:N2 | Laboratory reference strain. |

*Continued on next page*

*Continued*

| Resource | Designation | Source or reference | Identifiers | Additional information |
|---|---|---|---|---|
| Strain (*C. elegans*) | DA609 | *Caenorhabditis* Genetics Centre | RRID:WB-STRAIN:DA609 | Genotype: *npr-1(ad609)X.* |
| Strain (*C. elegans*) | OMG2 | this paper | | Genotype: *mIs12 [myo-2p::GFP]II; npr-1(ad609)X.* Originated from CB5584 and DA609. |
| Strain (*C. elegans*) | OMG10 | this paper | | Genotype: *mIs12 [myo-2p::GFP]II.* Originated from CB5584; outcrossed 6x to CGC N2. |
| Strain (*C. elegans*) | OMG19 | this paper | | Genotype: *rmIs349 [myo3p::RFP]; npr-1(ad609)X.* Originated from AM1065 and DA609. |
| Strain (*C. elegans*) | OMG24 | this paper | | Genotype: *rmIs349 [myo3p::RFP].* Originated from AM1065; outcrossed 6x to CGC N2. |
| Strain (*C. elegans*) | DR476 | *Caenorhabditis* Genetics Centre | RRID:WB-STRAIN:DR476 | Genotype: *daf-22(m130)II.* |
| Strain (*C. elegans*) | AX994 | Mario de Bono (MRC Laboratory of Molecular Biology) | | Genotype: *daf-22 (m130)II; npr-1(ad609)X.* |
| Software | Tierpsy Tracker (v 1.3) | *Javer et al., 2018* | | Software available at ver228. github.io/tierpsy-tracker. |
| Software | wormTracking Analysis | this paper | | Software available at github.com/ljschumacher/wormTrackingAnalysis. |
| Software | sworm-model | this paper | | Software available at github.com/ljschumacher/sworm-model. |

## Animal maintenance and synchronization

*C. elegans* strains used in this study are listed in Key Resources Table above. All animals were grown on *E. coli* OP50 at 20°C as mixed-stage cultures and maintained as described (*Brenner, 1974*). All animals used in imaging experiments were synchronized young adults obtained by bleaching gravid hermaphrodites grown on *E. coli* OP50 under uncrowded and unstarved conditions, allowing isolated eggs to hatch and enter L1 diapause on unseeded plates overnight, and re-feeding starved L1's for 65–72 hr on OP50.

## Bright field high-number swarming imaging

The strain used here (*Figure 1A* and *Video 1*) is DA609. On imaging day, synchronized adults were collected and washed in M9 buffer twice before several hundred animals were transferred to a seeded 90 mm NGM plate using a glass pipette. After M9 is absorbed into the media, ten-hour time-lapse recordings were taken with a Dino-Lite camera (AM-7013MT) at room temperature (20°C) using the DinoCapture 2.0 software (v1.5.3.c) for maximal field of view. Two independent replicates were performed.

## Bright field standard swarming imaging

Step-by-step protocol is available at dx.doi.org/10.17504/protocols.io.vybe7sn. All recordings from this dataset are listed in *Supplementary file 2*.

The strains used here (*Figure 1B*) are DA609 and N2. Prior to collecting the full dataset, a single batch of OP50 was grown overnight, diluted to $OD_{600}$ = 0.75, aliquoted for use on each imaging day, and stored at 4°C until use. Imaging plates were 35 mm Petri dishes containing 3.5 mL low pep-tone (0.013% Difco Bacto) NGM agar (2% Bio/Agar, BioGene) to limit bacteria growth. A separate batch of plates was poured exactly seven days before each imaging day, stored at 4°C, and dried at 37°C overnight with the agar side down before imaging. The center of an imaging plate was seeded with a single 20 µL spot of cold diluted OP50 one to three hours before imaging. The overnight plate drying step allowed the bacteria to quickly dry atop the media in order to achieve a more uni-form lawn by minimizing the 'coffee ring' effect that would thicken the circular edge of the bacterial lawn. For each imaging day, synchronized young adults were collected and washed in M9 buffer twice before 40 animals were transferred to a seeded imaging plate using a glass pipette.

Imaging commenced immediately following animal transfer in a liquid drop, on a custom-built six-camera rig equipped with Dalsa Genie cameras (G2-GM10-T2041). Seven-hour recordings with red illumination (630 nm LED illumination, CCS Inc) were taken at 25 Hz using Gecko software (v2.0.3.1), whilst the rig maintained the imaging plates at 20°C throughout the recording durations. Images were segmented in real time by the Gecko software. The recordings were manually truncated post-acquisition to retain aggregation and swarming dynamics only. The start time was defined as the moment when the liquid dried and the all the worms crawled out from the initial location of the drop, and the end time was when the food was depleted and worms dispersed with increased crawl-ing speed. Twelve independent replicates were performed for each strain.

## Bright field big patch swarming imaging

Step-by-step protocol is available at dx.doi.org/10.17504/protocols.io.vyhe7t6. All recordings from this dataset are listed in *Supplementary file 2*.

The experiments here (*Figure 1—figure supplement 1*) are identical to those in the bright field standard swarming imaging, except for two differences. First, the imaging plates were seeded with a 75 µL spot of diluted OP50 ($OD_{600}$ = 0.38) and allowed to inoculate overnight at room tempera-ture before being used for imaging the next day. Second, recordings were taken over 20 hr instead of seven. Eight independent replicates were per-formed for each strain.

## Bright field pheromone imaging

Step-by-step protocol is available at dx.doi.org/ 10.17504/protocols.io.vyie7ue. All recordings from this dataset are listed in *Supplementary file 2*.

The strains used here (*Figure 3—figure sup-plement 1*) are DA609, N2, DR476, and AX994. Bacteria aliquots and imaging plates were pre-pared as in the bright field standard swarming imaging assay. For each imaging day, synchro-nized young adults were collected and washed in M9 buffer twice before 40 animals were trans-ferred to a seeded imaging plate using a glass pipette. After M9 was absorbed into the media following worm transfer in liquid, imaging plates containing the animals were subjected to a gen-tle vibration at 600 rpm for 10 s on a Vortex Genie two shaker (Scientific Industries) to dis-perse animals and synchronize aggregation start across replicates. Imaging commenced 20 s after

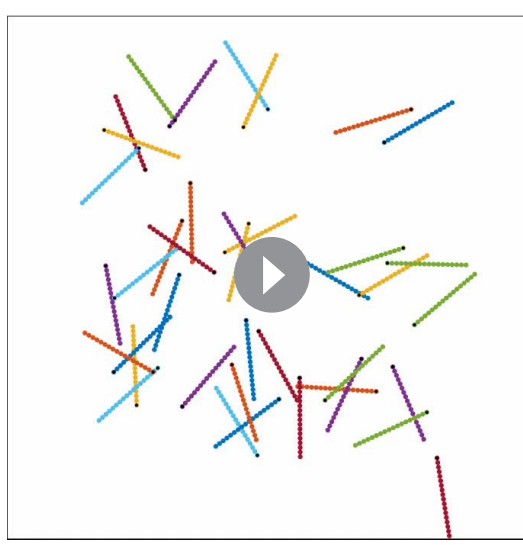

**Video 7.** Sample swarming simulation describing *npr-1* mutants. Background color shows relative food concentration with white indicating high food and black indicating no food. The video plays at 30x the normal speed.

DOI: https://doi.org/10.7554/eLife.43318.025

the vibration finish, using the same rig set-up as swarming imaging above, except one-hour recordings were taken. Images were segmented in real time by the Gecko software. At least eight independent replicates were performed for each strain. Automated animal tracking was performed post-acquisition using Tierpsy Tracker software (http://ver228.github.io/tierpsy-tracker/, v1.3), which we developed in-house (*Javer et al., 2018*). Images with were tracked with customized parameters to create centroid trajectories, 49-point worm skeletons, and a battery of features.

## Fluorescence aggregation imaging

Step-by-step protocol is available at dx.doi.org/10.17504/protocols.io.vzje74n. All recordings from this dataset are listed in *Supplementary file 2*.

The strains used here (*Figure 2*, *Videos 2–4*) are OMG2, OMG10, OMG19, and OMG24. One-color imaging consisted of pharynx-GFP labeled worms only, whereas two-color imaging also included a small number of body wall muscle-RFP labeled worms that were recorded simultaneously on a separate channel (thus readily segmented from the rest of the worms). The latter was necessary to follow individuals over a long period of time, particularly while inside a cluster, as frequent pharynx collisions inside clusters lead to lost individual identities and broken trajectories. For two-color imaging, animals with different fluorescent markers were mixed in desired proportion (1–3 red animals out of 40 per experiment) during the washing stage before being transferred together for imaging.

The data collection paradigm was identical to the bright field pheromone imaging assay in terms of bacteria aliquots, imaging plate preparation, and vibration implementation following animal transfer. The difference is that image acquisition was performed on a DMI6000 inverted microscope (Leica) equipped with a 1.25x PL Fluotar objective (Leica), a TwinCam LS image splitter (Cairn) with a dichroic cube (Cairn), and two Zyla 5.5 cameras (Andor) to enable simultaneous green-red imaging with maximal field of view. One-hour recordings were taken with constant blue (470 nm, 0.8A) and green (cool white, 1.4A) OptoLED illumination (Cairn), and images were acquired with 100 ms exposure at 9 Hz using Andor Solis software (v4.29.30005.0). The microscopy room was maintained at 21° C throughout the recording durations. Ten or more independent replicates were performed for each strain. We were able to reproduce stereotyped aggregation dynamics across replicates under our experimental paradigm (*Figure 1—figure supplement 2*). Image segmentation and automated animal tracking was performed post-acquisition using Tierpsy Tracker software (v1.3) with customized parameters, to create centroid trajectories, obtain two-point skeleton from pharynx-labeled individuals and 49-point midline skeletons from body wall muscle-marked ones, and extract various features. For body wall muscle-marked animals, trajectories were manually joined where broken due to tracking errors.

## Fluorescence aggregation tracking data analysis

The code for tracking data analysis is available at https://github.com/ljschumacher/wormTrackingAnalysis (*Schumacher et al., 2019*; copy archived at https://github.com/elifesciences-publications/wormTrackingAnalysis).

Tracked blobs were filtered for minimum fluorescence intensity and maximum area, to exclude any larvae and tracking artifacts, respectively, which appeared on the occasional plate. Local worm densities around each individual were calculated using $k$-nearest neighbor density estimation, where the density is $k$ divided by the area of a circle encompassing the $k$-th nearest neighbor. We chose $k = 6 \approx \sqrt{N}$ and verified based on visual assessment that the overall distribution of local densities changes very little with increasing $k$.

Reversals were detected based on a change of sign of speed from positive to negative, which was calculated from the dot-product of the skeleton vector (of the pharynx) and the velocity vector, and smoothed with a moving average over half a second. We only counted reversals that were at least 50 μm in length, and that moved at least half a pixel per frame before and after the reversal. Reversal events thus detected where binned by their local density. For each density bin, reversal rate was estimated as the number of events divided by the time spent in forward motion for that bin. The variability was estimated using a subsampling bootstrap: the reversal rate was estimated 100 times, sampling worm-frames with replacement, and estimating mean and standard deviation.

Speed profiles were generated by binning the measured speed values for local density, and then creating a histogram of speed values for each density bin.

Summary statistics of aggregation, such as pair-correlation and hierarchical clustering, where calculated as described in Appendix 1.

## Acknowledgements

We thank Camille Straboni for her contributions in the early stages of this project, Mario de Bono for his gift of strain AX994, Richard Morimoto for his gift of strain AM1065, Chad Whilding and Dirk Dormann for help with setting up the imaging system, Ivan Croydon Veleslavov for contributing to the code for parameter inference, Jochen Kursawe for helpful discussions, and Suhail Islam for computational support. Some strains were provided by the CGC, which is funded by NIH Office of Research Infrastructure Programs (P40 OD010440). This work has made use of the resources provided by the Imperial College Research Computing Service (DOI: 10.14469/hpc/2232) and the Edinburgh Compute and Data Facility (ECDF) (http://www.ecdf.ed.ac.uk/).

## Additional information

### Funding

| Funder | Grant reference number | Author |
| --- | --- | --- |
| Biotechnology and Biological Sciences Research Council | BB/N00065X/1 | Robert G Endres<br>André EX Brown |
| Medical Research Council | MC-A658-5TY30 | André EX Brown |

The funders had no role in study design, data collection and interpretation, or the decision to submit the work for publication.

### Author contributions

Siyu Serena Ding, Linus J Schumacher, Conceptualization, Data curation, Software, Formal analysis, Validation, Investigation, Visualization, Methodology, Writing—original draft, Project administration, Writing—review and editing; Avelino E Javer, Data curation, Software; Robert G Endres, Conceptualization, Resources, Formal analysis, Supervision, Funding acquisition, Investigation, Methodology, Writing—original draft, Project administration, Writing—review and editing; André EX Brown, Conceptualization, Resources, Software, Supervision, Funding acquisition, Investigation, Methodology, Writing—original draft, Project administration, Writing—review and editing

### Author ORCIDs

Siyu Serena Ding http://orcid.org/0000-0002-8590-3908
Linus J Schumacher http://orcid.org/0000-0003-0797-3406
Robert G Endres http://orcid.org/0000-0003-1379-659X
André EX Brown http://orcid.org/0000-0002-1324-8764

### Decision letter and Author response

Decision letter https://doi.org/10.7554/eLife.43318.033
Author response https://doi.org/10.7554/eLife.43318.034

## Additional files

### Supplementary files

• Supplementary file 1. Model parameters.
DOI: https://doi.org/10.7554/eLife.43318.026
• Supplementary file 2. Imaging datasets used in this study.
DOI: https://doi.org/10.7554/eLife.43318.027
• Transparent reporting form

DOI: https://doi.org/10.7554/eLife.43318.028

**Data availability**

All data generated and analysed during this study is deposited on the Open Worm Movement Database community page (https://zenodo.org/communities/open-worm-movement-database/). Each recording has a separate DOI, which can be found in Supplementary file 2. The code for model simulations is available at https://github.com/ljschumacher/sworm-model (copy archived at https://github.com/elifesciences-publications/sworm-model).

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

# Appendix 1

DOI: https://doi.org/10.7554/eLife.43318.029

## 1 Agent-based simulations

We aim to create a model of worm locomotion and interaction that recapitulates aggregation and swarming behavior. Many mechanical models of worm locomotion exist in the literature, but we aim for a simpler representation of each individual worm, so that computationally inexpensive simulations of tens to hundreds of worms allow rapid hypothesis exploration and testing.

### 1.1 SPP worm model

Each agent is represented by $M$ nodes connected linearly by $M-1$ segments. Each node moves as a self-propelled particle with a preferred speed $v$. At each time-step, the direction of movement is updated based on phenomenological forces representing active movement, interactions with other worms, and constrains to ensure the worm does not extend in length or bend excessively. Nodes follow forces in the over-damped regime, $\mathbf{v} \sim \mathbf{F}$, with periodic boundary conditions.

The code for model simulations is available at github.com/ljschumacher/sworm-model (**Schumacher, 2019**; copy archived at https://github.com/elifesciences-publications/sworm-model).

### 1.1.1 Self-propelled movement

The self-propulsion is modeled as a motile force, $\mathbf{F}_{m_1}^{t+1} = v\big[\cos(\phi_1^{t+1}),\ \sin(\phi_1^{t+1})\big]$, on node 1, that is the head node. Note that for notational convenience we ignore the constant of proportionality, implicitly writing $F = \tilde{F}/\gamma$, where $\tilde{F}$ has units of force and $F$ has units of velocity.

To mimic a worm's persistent movement with directional changes over time (**Salvador et al., 2014**), we add a stochastic contribution to the head node's movement, given by $\phi_1^{t+1} = \phi_1^t + \eta\xi$, where $\phi_i$ is the orientation of node $i$ with respect to the $x$-axis, $\eta$ is the noise strength, and $\xi$ is a normally distributed random variable. The noise is parameterized by analyzing the directional auto-correlation of single worm simulations, and set so that the autocorrelation after 25 s (roughly the time it takes an npr-1 worm to cross the 8.5 mm food patch) is, on average, less than 0.23. This value is equivalent to a random reorientation between $-3\pi/4$ and $3\pi/4$, and thus reflects that over a distance equivalent to the food patch size, worms should lose all memory of their orientation. For N2 simulations, which move at a lower speed, the noise strength is scaled by a factor of $\sqrt{v_{npr-1}/v_{N2}}$, which results in the same condition.

For the nodes following the head node, the direction of movement is given by the tangent vector towards the next node. For node $i$, the tangent vector is calculated as $\mathbf{s}_i = [(\mathbf{x}_i - \mathbf{x}_{i+1}) + (\mathbf{x}_{i-1} - \mathbf{x}_i)]/2$, that is the average between the direction towards the previous node and the direction from the next node. The motile force on node $i$ is then given by $\mathbf{F}_{m_i}^{t+1} = v\mathbf{s}_i$.

After forces have been applied and the nodes' positions updated, the headings are updated to reflect the direction of the displacement for calculating the movement in the next time step.

### 1.1.2 Undulations

To mimic more worm-like movement (**Figure 6—figure supplement 2H**), we impose a sinusoidal contribution to the direction of the head node's movement. If $\theta$ is the direction of movement in the worm's reference frame, and $\phi_i$ the orientation of node $i$ with respect to the $x$-axis, we assume the heading of the worm internally oscillates with angular frequency $\omega$ and amplitude $\theta_0$, so that

$$\theta(t) = \theta_0 \sin \omega t. \tag{1}$$

This prescribes the change in direction for the head node at every time step, such that

$$\phi_1^{t+1} = \phi_1^t + \Delta\theta^{t+1} + \eta\xi = \phi_1^t + \theta^{t+1} - \theta^t + \eta\xi, \tag{2}$$

where $\omega = 2\pi \times 0.6 Hz$, $\theta_0 = \pi/4$, and the $m$th node's internal oscillator is phase-shifted by $\Delta\Psi_m = 11.76 \times m/M$.

### 1.1.3 Taxis

To investigate the effect of taxis in our simulations, we treat the movement of the head node as an attracting walk with respect to other worm's nodes within an interaction radius $R_{\text{taxis}}$ (see *Hannezo et al., 2017*, SI). This was implemented as an additional term $f_t \mathbf{p}_{\text{taxis}}$ added to the motile force that affects its direction as well as its the magnitude (reflecting additive contribution from multiple neighboring worms). The parameter $f_{\text{taxis}}$ controls the strength of taxis per other worm. The taxis force is additionally weighted by $1/r$ to reflect that nearby neighbors exert a stronger attraction, that is as if mediated by a non-degrading, diffusible factor, such as oxygen or $CO_2$. The vector $\mathbf{p}_{\text{taxis}}$ is the sum of the directions towards other worms' nodes within the interaction radius, $R_{\text{taxis}}$, so that for worm $k$, the taxis contribution to the motile force is

$$\mathbf{p}_{\text{taxis},k} = \frac{1}{M}\sum_j \left[\delta(r_c \leq r_{jk} \leq R_{\text{taxis}})\frac{r_c}{r_{jk}} - \delta(r_{jk} < r_c)\right]\frac{\mathbf{x}_j - \mathbf{x}_k}{|\mathbf{x}_j - \mathbf{x}_k|}. \tag{3}$$

The sum is over all nodes of other worms, and the force is normalized by $M$ to make it independent on the number of nodes in a worm. To prevent excessive overlap of worms, the taxis force become repulsive for worms that overlap, hence the negative second term.

### 1.1.4 Length constraints

To enforce approximately constant length of the worm, each node is connected by non-linear springs of rest length $l_0$ that resist an extension $\delta l = l - l_0$, where $l$ is the length of the segment, with opposing force

$$\mathbf{F}_l = k_l\hat{\mathbf{l}}\frac{\delta l}{1 - \left(\frac{\delta l}{l}\right)^2}, \tag{4}$$

which points along the direction of the segment, $\hat{\mathbf{l}} = \mathbf{l}/l$.

### 1.1.5 Volume exclusion

For supplementary simulations with volume exclusion (*Figure 6—figure supplement 5B*), the forces are modified as follows when two nodes are overlapping: Any two nodes $i$ and $j$ of different objects that are closer than $2r_c$ exert contact forces onto each other (nodes within the same object can overlap without contact forces). The total force acting on node $i$, $\mathbf{F}_i$ is projected onto the connecting line between the nodes, and if this projected force is pointing towards node $j$ (pushing rather than pulling), it is added to $\mathbf{F}_j$. The contact force of $j$ onto $i$ is calculated mutatis mutandis.

### 1.1.6 Adhesion

To assess how aggregation is affected by a moderate adhesion (equal to both strains), such as could arise through liquid film forces (*Gart et al., 2011*), we implemented a soft-core version of the Lennard-Jones potential. This gives rise to a force between any two nodes of *different* worms that is repulsive at short distances, attractive at intermediate distances, and zero at long distances. The force between two nodes separated by $r < 3.75r_c$ (the cut-off was chosen to limit adhesive force to nearest neighbors) is given by a soft-core potential of a generalized Lennard-Jones form (*Heyes, 2010*):

$$F_a = 8\frac{\epsilon_a}{\tilde{r}}\left[\left(\frac{\sigma_a}{\tilde{r}}\right)^2 - \frac{\sigma_a}{2\tilde{r}}\right], \tag{5}$$

where $\tilde{r} = 2\sigma_a/3 + r$. The parameter $\sigma_a = 2r_c$ was chosen so that the force becomes attractive at a distance greater than the node particle size, the exponent of the attractive term was chosen as -1 to reflect the $1/r$ dependence estimated for liquid film tension between two worms (*Gart et al., 2011*), and the exponent of the repulsive term was set as -2 to win over the attractive term at short distances (to ensure volume exclusion). Note that adhesion is not used in any of the results of this work and was only used to illustrate its unrealistic effects on aggregation (*Figure 5D*).

### 1.1.7 Switching between slow and fast movement

Worms stochastically switch between movement at speeds $v_0$ and $v_s$ with rates that depend on the local density of worms surrounding them. In the absence of other worms, the (Poisson) rates are $k_{s0}$ to slow down from $v_0$ to $v_s$, and $k_{f0}$ to speed up from $v_s$ to $v_0$. These rates increase and decrease, respectively, with the number of neighboring worm nodes within $r_i$ of any node of the worm, such that

$$k_{\text{slow}} = k_{s0} + k_s'\rho, \tag{6}$$

where the linear dependence is chosen for simplicity, and $k_s'$ is a free parameter, and

$$k_{\text{fast}} = k_{f0}\exp\left[-k_f'\rho\right], \tag{7}$$

where the exponential decay with decay constant $k_f'$ was chosen to provide a lower bound of 0 for the rate. Note that the rate of switching to fast movement is related to the duration of a period of slow movement via $\tau_{\text{slow}} = 1/k_{\text{fast}}$ (for Poisson rates).

The local density $\rho$ is estimated by counting the average number of other worms' nodes in a radius $r_i$ around each node of the current worm.

$$\rho = \frac{1}{M}\sum_m^M\sum_n^N\sum_j^M\Theta(r_i - |r_m - r_{nj}|), \tag{8}$$

where $|r_m - r_{nj}|$ is the distance from the current node $m$ to node $j$ of worm $n$, $\Theta$ is the Heaviside step function (such that $\Theta(x) = 1$ if $x>0$), and the sum over other worms skips the index of the current worm.

For simulations with undulations, when a worm has slowed down to $v_s$, the angular frequency of its internal oscillators slows down accordingly to $\omega_s = \omega v_s/v_0$.

### 1.1.8 Reversals

To model reverse movement, we switch the direction of the nodes for the duration of the reversal, such that movement originates from the tail and the rest of the body follows. Reversals events are generated stochastically, with Poisson-rate $r_{\text{rev}}$, which depends on the local density via

$$r_{\text{rev}} = r'\rho,, \tag{9}$$

where $r'$ is a free parameter, and $\rho$ is the local density as estimated above. Once a reversal rate has started, it lasts for $t_{\text{rev}} = 2s$, unless otherwise aborted (see Contact-dependent reversal events).

### 1.1.9 Reversals with undulations

Upon reversals, we have also to reset the phase of the internal oscillator prescribing the undulating movement of the worm to match its current shape (as the phase may have decoupled from the shape during movement). Recall that the internal orientation of a node with index $i = s/L_w$, where $s$ is the arc-length along the worm, is changing with the node's internal oscillator according to

$$\theta = \theta_0 \sin\left(\omega t - s\frac{\Delta\psi}{l}\right),$$

and the derivative with respect to arc length, $s$, differentiating towards the head, that is decreasing $s$, gives

$$-\frac{\mathrm{d}\theta}{\mathrm{d}s} = \frac{\Delta\psi}{l}\theta_0 \cos\left(\omega t - s\frac{\Delta\psi}{l}\right).$$

Both the angle, $\theta$, and the curvature, $\frac{\mathrm{d}\theta}{\mathrm{d}s}$, are needed to estimate the phase uniquely, using

$$\frac{\theta}{-\frac{\mathrm{d}\theta}{\mathrm{d}s}\frac{l}{\Delta\psi}} = \tan\left(\omega t - s\frac{\Delta\psi}{l}\right),$$

which we re-arrange to get the phase, that is the node's oscillator's internal time,

$$\psi = \omega t - s\frac{\Delta\psi}{l} = \arctan\left(\frac{\theta}{-\frac{\mathrm{d}\theta}{\mathrm{d}s}\frac{l}{\Delta\psi}}\right). \tag{10}$$

We use this expression to set the phase of the head/tail node after a reversal starts/ends, and set the phase of the rest of the worm according to $\psi_i = \psi - i\Delta\psi$.

### 1.1.10 Contact-dependent reversal events

The rate of reversal events depends on whether the head and tail are in close proximity with other worms, being $r_{\mathrm{rev}}$ when only the head or tail is in close proximity to another worm, but not both, and zero otherwise. Head and tail nodes are specified as the first and last 10 percent of the nodes (rounded), respectively. Contact is registered if any other worm's nodes are within $r_i$ of the head/tail nodes. If the worm is going forward and the tail is in contact, but the head is not, reversals occur with rate $r_{\mathrm{rev}}$. If the worm is already reversing, and the tail is not in contact, but the head is, reversals stop with the same rate. If both or neither head and tail are in contact, no reversals occur (adding reversal rates as measured for freely moving worms did not qualitatively change the aggregation outcome of simulations).

### 1.1.11 Adaptive time-step

The time-step of simulations is chosen adaptively to maintain accuracy at higher forces. To achieve this the time-step scales inversely with the maximum magnitude of forces in the system, $\mathrm{d}T \sim \mathrm{d}T_0/\max(F_i)$. The precise scaling is chosen so that the node with the highest force acting on it moves no further in one time-step than $1/2$ of the node radius.

## 1.2 Food depletion

For simulations with food depletion, food is initialized uniformly on a grid of size $L/(4r_c)$, where $r_c$ is the node radius. Food concentration is set equal to 100 in arbitrary units. Before worm movement is calculated, food concentrations are checked. If the food is depleted at the grid-point closest to the head node of a worm, the worm moves at the faster speed $v_0$, regardless of other interactions (i.e. does not slow down and speeds up if previously slowed down). After worm movements, food is consumed in each grid-point by an amount $r_{\mathrm{feed}}$ per worm-head in that grid-point, with a minimum of zero food.

## 2 Parameter inference

### 2.1 Inference scheme

We employ approximate Bayesian inference with rejection sampling (***Beaumont et al., 2002***; ***van der Vaart et al., 2016***). We sample from our prior distribution of the parameters (see Reduction to feasible parameter space) and run simulations for these samples. Similarity to the experimental data is then computed based on summary statistics (see Summary statistics), and the closest fraction $\alpha$ of the simulations are chosen. To estimate the posterior distribution from

these chosen parameter samples, we construct a kernel density estimation, with the weight for each sample chosen inversely proportional to the distance from the experimental data.

## 2.2 Reduction to feasible parameter space

For the four-parameter model, with density-dependent reversals ($r'$), speed-switching rates ($k_s', k_f'$) and taxis interactions ($f_t$), we employ a strategy to exclude unfeasible regions of parameter space before running long simulations. Our reasoning is that interactions must be such that pairs of worms should not be stable for long times, and cluster of worms should be stable/unstable for npr-1/N2. We first sample parameters for pilot simulations from a regular grid, with $10^d$ samples, where $d$ is the dimensionality of our parameter space. We then run simulations of worms starting as an overlapping pair, and assess whether they are within 1 mm of each other after 1 min of simulation (taking the median of 10 repeated simulations). If their separation is below the threshold, we discard the parameter sample. The remaining parameter samples are used to run simulations in which worms start out in a cluster (by confining their initial positions to a circle of 1.8 mm radius). These simulations are run for 300 s, after which stability of the cluster is assessed by calculating the radius of gyration of the head-nodes of the worms. If the radius of gyration is above 3 mm (which corresponds approximately to worms being uniformly distributed within a square of 7.5 mm side length), the cluster is deemed not stable and the parameter sample is discarded for npr-1 simulations, and kept for N2 simulations. Both the pair- and cluster-stability thresholds are chosen conservatively to include rather than exclude potential parameter samples. Never the less, only a few percent of the initial parameter space remain as feasible for further inference. The remaining parameter samples are used to construct a prior distribution via kernel density estimation, that is centering a Gaussian distribution on each sample.

For the N2 parameterization, only pilot runs with $f_{\text{taxis}} = 0$ were accepted, so we chose to sample this parameter on a $\log_{10}$-scale for both strains. When constructing the approximate posterior distribution this change in prior $\pi$ was taken into account by weighting each sample with the appropriate importance factor of $\pi_{\text{new}}/\pi_{\text{old}}$.

## 2.3 Summary statistics

We use the following summary statistics to quantify aggregation and compute the similarity between simulations and the experimental data:

1. The pair-correlation function compares the density of neighbors at a distance $r$ to that expected under a uniform random distribution (**Gurry et al., 2009**):

$$S_1 = g(r) = \frac{A}{N(N-1)} \frac{\sum_i^N \sum_{j\neq i}^N \mathbf{1}_{ij}(r - a < r_{ij} \leq r)}{\pi(r^2 - (r-a)^2)}, \tag{11}$$

where $r_{ij}$ is the distance between objects $i$ and $j$, $A = L^2$ is the size of the simulation domain, chosen to match the estimated are of the food patch in experiments.

2. Hierarchical clustering (as implemented in Matlab's linkage function) quantifies the structure of a point pattern through agglomerative clustering. Each frame results in a dendrogram, or clustering tree. We summarize the distribution of these clustering trees through the overall distribution of branch lengths, $S_2$.

3. The standard deviation of the positions, $\sigma(\mathbf{x}) = \sqrt{\sigma(x)^2 + \sigma(y)^2}$, is a simple way to quantify the spread of points $\mathbf{x} = (x, y)$, which we average over time to give

$$S_3 = \langle \sigma(\mathbf{x}) \rangle_t. \tag{12}$$

4. The kurtosis or the sharpness of the distribution of positions,

$$S_4 = \langle \mathrm{Kurt}(\mathbf{x}) \rangle_t. \tag{13}$$

To compute these summary statistics, we randomly sample frames from experiments and simulations such that on average we have one frame every three seconds. To mimic the partial information about a worm's position obtained from the pharynx-labelled imaging, we restricted the simulation analysis to the first 16 percent of the nodes (based on measurements of pharynx size relative to worm body length), from which centroid positions for each worm were obtained. We also computed the nematic order parameter (*Weitz et al., 2015*), but found these to be low ($\approx 0.2$) for both strains, and hence not an informative summary statistic of aggregation in our system.

Note that when calculating summary statistics for simulation outputs, periodic boundary conditions have to be taken into account. This means calculating any distances $r$ as $\min(|r|, |L - r|)$, and furthermore calculating the mean positions, $\bar{x}_i$, in dimension $i$ (used in $S_3$ and $S_4$) as

$$\bar{x}_i = \frac{L}{2\pi}(\mathrm{atan2}(-\bar{s}_{x_i}, -\bar{c}_{x_i}) + \pi), \tag{14}$$

where $s_{x_i} = \sin(x_i/(2\pi L))$, $c_{x_i} = \cos(x_i/(2\pi L))$ and $\mathrm{atan2}$ is the four-quadrant inverse tangent.

### 2.3.1 Distance function

Before combining the summary statistics into a single distance function, we scale them for their overall magnitude and dimensionality as follows: We take the log-ratio of the summary statistics from experiments and simulations (*Barnes et al., 2012*) to adjust both for the different scale of bins within distributions, and the different scales of summary statistics overall, such that each statistic is weighted approximately equally, irrespective of its average magnitude.

We further note that higher dimensional summary statistics result in larger distance values, even if the difference in each dimension is equal to that of a lower dimensional statistic. We choose to normalize for this by dividing the distance by the square root of the dimensionality.

Thus, our distance function for summary statistic $S_i$ with dimensionality $D_i$ is given by

$$d_i = ||\log S_{i,\mathrm{obs}} - \log S_{i,\mathrm{sim}}||_2 / \sqrt{D_i}. \tag{15}$$

Using $\log$-ratios can cause infinite distances if any of the $S_{i,\mathrm{sim}} = 0$. To avoid this, we cap the simulation data 0.005, that is we set $S_{i,\mathrm{sim}} = \max(S_{i,\mathrm{sim}}, 0.005)$. This limits the penalizing effect of empty bins and the tails of a distribution on the overall distance function.

### 2.3.2 Alternative weighting of summary statistics

We explored optimizing the weighting of summary statistics to maximize the distance between our prior and posterior distribution over the parameters (*Harrison and Baker, 2017*), but this led to weighting of the summary statistics (with all the weight concentrated in $S_2$ and $S_3$) which did not match with visual inspection of the closest matching simulations. In other words, equally weighting all summary statistics returned simulations that better reflected our intuition for what constitutes a good match, in particular for the npr-1 parameterization. In the interest of completeness we describe here the method of *Harrison and Baker (2017)* applied to our data, as it informed our thinking, even though we did not use the results.

To try and optimize the weighting of our summary statistics, we optimized the Hellinger distance between our prior and posterior distribution over the parameters (*Harrison and Baker, 2017*), with weak regularization ($\lambda = 10^{-4}$) of the parameters included in the objective function. Distributions are calculated using kernel density estimation as described above, and as an optimization procedure we use the genetic algorithm provided in Matlab's global optimization toolbox. With the weightings $w_i$ for each summary statistic thus optimized, the overall distance is $d = \sum_i w_i d_i$.

This method of adaptively weighting summary statistics is still sensitive to the choice of statistics. Our choices are by no means exhaustive, and we chose to focus on statistics commonly used to quantify aggregation (pair-correlation function and hierarchical clustering) and the shape of distributions (variance and kurtosis).

To ensure that the same summary statistics are chosen for the parameter inference for either strains, we jointly optimize the posterior distribution for both strains, by minimizing the objective function $L = -(H_1 + H_2)$, where $H_i$ is the Hellinger distance between the prior and the posterior for strain $i$.

## 2.4 Kernel density estimation

For plotting the marginal joint distributions between pairs of parameters, we use ksdensity (Matlab, R2018a). For constructing the higher-dimensional parameter distributions to sample from, we implement the kernel density estimation using `gmdistribution` (Matlab, R2018a) with Silverman's rule of thumb for the bandwidths.

## 2.5 Sampling sequence

We first sampled 100,000 samples from our prior, resulting in 11,214 simulations for npr-1 and 1394 simulations for N2 (only a fraction of parameter samples resulted in full simulations because samples resulting in stable pairs and stable/unstable clusters were rejected for N2/npr-1, see Reduction to feasible parameter space). To improve the successful sampling rate, we constructed an approximate posterior distribution from the initial samples, and continued sampling from this posterior distribution, thus ensuring the samples were concentrated in the appropriate regions of parameter space. This change in the sampling distribution was accounted for when constructing the final posterior (*Figure 6D*) distributions through weighting by the ratio of the initial prior distribution to the proposal distribution (with a small regularization to avoid division by near-zero weight for outlier samples). In this second round of sampling we generated 13,341 simulations for N2 and 27,384 samples for npr-1.

# Appendix 2

DOI: https://doi.org/10.7554/eLife.43318.029

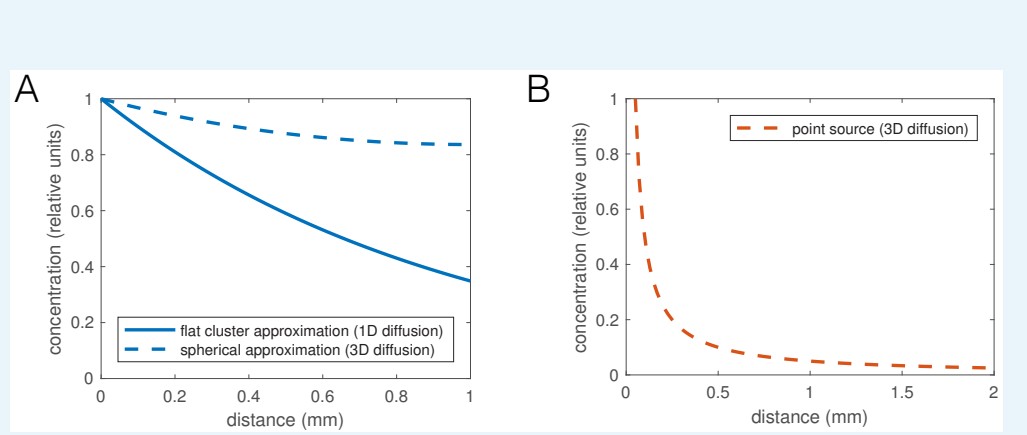

**Appendix 2—figure 1.** Oxygen consumption-diffusion calculations predict shallow $O_2$ concentration gradients. (**A**) Plot of feasible oxygen gradients inside worm aggregates. The oxygen concentration decays with length constant $\sqrt{D/\mu} \approx 1\,\mathrm{mm}$, with diffusion constant $D \approx 2.1 \times 10^{-5}\,\frac{\mathrm{cm}^2}{s}$ (in water) and oxygen consumption rate $\mu \approx 0.14\,\mathrm{min}^{-1}$ (estimated as an upper bound for 200 pl/min [*Shoyama et al., 2009*; *Suda et al., 2005*] at 21% oxygen and 8000 pl worm volume). The thinnest dimension of a cluster is relevant for diffusion, which is its thickness. We can approximate the cluster geometry either as flat, which results in a 1D diffusion gradient (solid line), or as hemispherical, which we approximate by spherically symmetric diffusion in 3D (dashed line). In either case the reaction-diffusion equation $\frac{\partial c}{\partial t} = D\Delta^2 c - \mu c$ was solved at steady state. (**B**) Gradient of diffusible, non-degrading signal, $\partial t$ for example $CO_2$, outside a point source. Without decay, this problem is equivalent to calculating the $\lambda$ potential around a point charge, and the concentration would be $c = \frac{\lambda}{4\pi Dr}$, in 3D, where $\lambda$ is the production rate times the volume of a worm, 0.14/min (equal and opposite to the $O_2$ consumption, based on mass conservation). A point source represents the contribution of a short section of a worm, and the contributions of many worms in an aggregate would integrate to give an approximately logarithmic gradient of signal outside the aggregate.
DOI: https://doi.org/10.7554/eLife.43318.031

