## [Decision Letter]

Thank you for submitting your article "Shared behavioral mechanisms underlie *C. elegans* aggregation and swarming" for consideration by *eLife*. Your article has been reviewed by two peer reviewers, and the evaluation has been overseen by a Reviewing Editor and Naama Barkai as the Senior Editor. The following individuals involved in review of your submission have agreed to reveal their identity: Jonathan Hodgkin and Oleg A Igoshin.

The reviewers have discussed the reviews with one another and the Reviewing Editor has drafted this decision to help you prepare a revised submission.

The manuscript of Ding et al. investigates collective feeding in the roundworm *C. elegans* with combination of time-lapse imaging, image analysis and mathematical modeling. It's a very well-written paper that presents broadly interesting and intriguing set of observation. Combination of experimental, quantitative analysis and modeling is the major strength of the paper. The authors study a prey-induced worm collective motility behavior called "swarming" and propose a mechanistic basis for this phenomenon. Both individual and population-level behaviors are quantified. Subsequent modeling is used to identify three key behaviors that drive aggregation: edge reversals, a density-dependent switches in speeds, and taxis towards neighboring worms. The same models can account for swarming when local food depletion is taken into account.

Major points:

1) Importantly, the modeling approach could be improved given the wealth of the experimental data. Specifically, it is surprising that the authors chose a fully phenomenological approach to construct their agent-based models that uses very little of the quantified data. Furthermore, authors do little to quantitatively compare resulting behaviors of agents in the model to those of tracked worms. Therefore, the reviewers are not fully convinced that identified behaviors in the model are either necessary or sufficient (given considerable noise in individual behaviors) for the observed population behaviors.

To circumvent this issue, measured individual worm behaviors should be used in constructing and/or constraining their model (see for example References PMID: 30514635, PMID: 28533367). Importantly, the noise and variability of the movement of agent match those of worms. It is not clear if this is the case in the current model. Furthermore, comparison of the resulting behaviors of agents and worms would allow testing the model. For example, is there correlation (e.g. between the agent velocity and the vector to nearby worm) that can quantify the degree of taxis? Can this correlation be compared between the model and experiments? Similar quantitative characteristics of other proposed 'key behaviors" need to be compared. Can changes in worm behaviors in nutrient-rich vs depleted regions be quantified and fit into the model? Either of the two approaches or some combination of them is needed to show convincingly that the postulated behaviors are indeed observed and responsible for the observed population self-organization.

2) The authors discuss possible neuronal mechanisms underlying their behavioural rules. Ideally, it would be good to apply or at least suggest possible experimental tests of these mechanisms, in particular of the proposed important medium-range taxis between neighbouring worms. It is suggested that this might be O2 tension mediated: how might this be tested? They do provide convincing experiments that exclude any significant contribution of ascaroside pheromones to swarming.

Other comments:

- The work also mainly compares behaviour in the standard lab strain N2, which is 'hypo-social', to behaviour in a 'hyper-social' npr-1 mutant of N2. Neither of these exactly corresponds to the natural state of almost all wild *C. elegans*, so the analysis is somewhat artificial; this might be more explicitly admitted.

- The authors state that to their knowledge "this swarming phenotype has not been reported in *C. elegans* previously", which betrays ignorance. For example, the following observation in Hodgkin and Barnes (2002, PMID: 1684664): "When a population has consumed almost all the bacteria, the worms… form a swarming mass that moves as a wave across the remaining bacterial lawn, and then disperses when all the bacteria have been consumed." This indicates that N2 swarms under appropriate conditions, perhaps of higher bacterial food density. The authors' experiments were performed only on "thin, even bacterial lawns".

- They also state that "it is still unclear whether aggregation and swarming have a function in the wild". Swarming is almost certainly naturally advantageous in predation on pathogenic or repellent bacteria, as a result of the 'wolfpack' effect that has been studied in myxobacterial swarms, to which some reference should be made. This reviewer, and others working on interactions between *C. elegans* and soil bacteria, have often observed N2 swarming at the edge of unpalatable bacterial lawns. These swarms appear to allow the worms to overcome bacterial defences by collective action of digestive enzymes or other secretions.

- Do worms in the experiments move in purely 2D, i.e. never crawl on top of one another and never overlap? Is this true for the agents in the simulations?

- Better justification of the level of complexity chosen to model worms with agent-based approach is necessary. Why would simple "vector particle" models fail? How important is that each worm is multi-segmented? How important are the elastic properties of the worm? Does the assumption of the propulsive force generation at the head vs throughout the body affects the motion?

- By comparison with the bacterial literature, the term swarming used by the authors could be somewhat confusing (e.g. by comparison with the widespread phenomenon referred to as bacterial swarming, e.g. PMID: 20694026). It is also probably acceptable to use the term "swarming" in the worm field, but a clear definition should be added in the Introduction.

---

## [Author Response]

Major points:1) Importantly, the modeling approach could be improved given the wealth of the experimental data. Specifically, it is surprising that the authors chose a fully phenomenological approach to construct their agent-based models that uses very little of the quantified data.

We thank the reviewers for their suggestions, however we believe that a phenomenological approach is more appropriate in our case. We lack a single-worm model that is accurate and simple enough for inference. We have recently shown that we can simulate accurate locomotion using a neural network (https://www.biorxiv.org/content/10.1101/222208v1), but achieving this accuracy requires hundreds of parameters. We feel a more appropriate level of description for modeling collective worm behavior is the persistent random walk, which has previously been shown to work well for single-worm foraging (Salvador et al., 2014 – now cited in Appendix 1 under “self-propelled movement”). At the level of this more tractable effective model, we thoroughly considered previously available single worm data to parametrize the size of our agents, rates of slowing, and the duration of slow movement periods at zero neighbor density. Other data on the frequency of undulations and worm shapes were ultimately not considered, as we did not find them relevant to the aggregation mechanism (see for example Figure 6—figure supplement 2H, where we add undulations).

Furthermore, authors do little to quantitatively compare resulting behaviors of agents in the model to those of tracked worms. Therefore, the reviewers are not fully convinced that identified behaviors in the model are either necessary or sufficient (given considerable noise in individual behaviors) for the observed population behaviors.

To address the question of necessity of the observed behaviors, we have run additional simulations and added Figure 6—figure supplements 2 and 3 in response to the reviewers’ suggestion. Starting from parameter values obtained from the mean of the posterior distribution as a reference, we removed or perturbed individual model behaviors. These simulations show that removing speed switching or taxis from the model disrupts aggregation, while removing reversals reduces the overall quantitative agreement with experimental data (Figure 6—figure supplement 2B-D). In some cases, removing individual model behaviors also produces a different correlation of velocity and orientation between neighbors from what we measure in experiments (Figure 6—figure supplement 3).

To circumvent this issue, measured individual worm behaviors should be used in constructing and/or constraining their model (see for example References PMID: 30514635, PMID: 28533367).

We appreciate the reviewers’ suggestions and have considered this approach. However, we favor a Bayesian inference approach instead of feeding the model at every time-step with observed tracking data (which in our case is imperfect), as other work on bacterial systems has done, including the suggested references. The inference approach not only gives us the most likely model parameters and their uncertainty, but also has several additional advantages, such as providing information on model identifiability or “sloppiness”, automatically adjusting for model complexity when sampling in higher dimensional parameter spaces, and being easily extensible to incorporate multiple sources of experimental data. In response to your suggestions, we now discuss the data-driven approach as an alternative (Discussion, fourth paragraph).

Importantly, the noise and variability of the movement of agent match those of worms. It is not clear if this is the case in the current model.

Noise and variability of movement have indeed already been considered when parameterizing the angular noise of movement relative to the size of food patch. In our experiments, long-term persistence of movement is disrupted when worms reach the edge of the food lawn. Thus the noise parameter was chosen to ensure an almost complete reorientation (velocity autocorrelation of 0.23, corresponding to choosing a random angle between ± 3/4π) after a time it takes a single worm to move across a food lawn (about 25 seconds for *npr-1* mutants moving at 330 μm/s). This is described in Appendix 1 under “self-propelled movement” and the velocity autocorrelation for our simulation is in Figure 6—figure supplement 4A3. To further compare whether the noise and variability of movement matches that of worms in our experiments, we analyzed the velocity autocorrelation in our worm tracking data. These show similar decay in velocity autocorrelation for individually tracked *npr-1* and N2 worms (Figure 6—figure supplement 4A1), and a stronger decay for *npr-1* worms in 40-worm experiments (Figure 6—figure supplement 4A2). The latter are thus interacting and expected to have less persistent movement as a result. We have further verified that the model behavior is robust to changing the noise parameter to as low as 0 and as high as 0.08 (which roughly corresponds to the autocorrelation measured for interacting worms in our experiments (Figure 6—figure supplement 4A2); higher noise values not tested), as can be seen in Figure 6—figure supplement 2E-G. These results show that aggregation in our model is robust to noise.

Furthermore, comparison of the resulting behaviors of agents and worms would allow testing the model. For example, is there correlation (e.g. between the agent velocity and the vector to nearby worm) that can quantify the degree of taxis? Can this correlation be compared between the model and experiments?

We have calculated the correlation of agent velocity and the vector to nearby worms to quantify the degree of taxis (Figure 6—figure supplement 3B). In our experiments, this correlation is nearly zero for all distances up to 2 mm, which is larger than the size of a typical worm cluster. This may not be intuitive, but the correlation was comparably low when we quantified the same statistic in our simulations, which have a low but non-zero taxis parameter. We suspect the reason is twofold: a) the taxis effect is only a small influence on the instantaneous direction of movement of a worm, compared to persistence and noise; and b) we only tracked the pharynx in our experiments, and reproduced this restriction in our analysis of simulations, but the whole body of the worm is likely giving relevant cues to any chemical or mechanical taxis. This brings with it the caveat that there is unseen worm density that affects any potential taxis behavior, but which remains undetectable in our tracking. Thus, our analysis shows that the attractive taxis is important for aggregation and may be non-zero (as it is in the parameterized simulations), even if it is difficult to detect with correlation analysis. Our modeling approach thus clearly predicts taxis, or a phenomenologically equivalent behavior, to be an important factor in aggregation, while our analysis also shows that this would have likely been missed by a purely data-driven approach.

Similar quantitative characteristics of other proposed 'key behaviors" need to be compared.

We now compare our inferred parameters for reversal and speed switching rates with our measured data (Figure 6—figure supplement 4B-C). The agreement of the reversal rate is imperfect, which may be in part due to the sensitivity of experimental analysis. When quantifying reversals, we had to make a choice as to when a reversal is substantial enough, for which we imposed a minimum absolute speed and a total path length of the reversal of 50 μm (roughly one worm width). Shorter reversals may more likely be due to noisy tracking or small head-movements. However, many reversals are not tracked from the beginning to the end, especially at higher densities; therefore any threshold on path length may falsely exclude partially tracked reversals.

To compare speed switching rates *k*_f_ and *k*_s_, we unfortunately cannot reliably measure these experimentally. However, we have quantified the ratio of worms moving at fast (up to 350 μm/s) versus slow (<100 μm/s for *npr-1*, <50 μm/s for N2) speeds in experiments and simulations at various local densities. The disagreement (Figure 6—figure supplement 4C) may indicate that the exponential form of *k_f_(ρ*) (see Figure 5C and main text) is not the best approximation. However, aggregation in the model is not sensitive to speed switching rates, as shown by the broad posterior distributions for the inferred parameters (Figure 6C-D), thus we leave it for future work to consider different functional forms of *k*_f_(*ρ*).

Can changes in worm behaviors in nutrient-rich vs depleted regions be quantified and fit into the model?

We have used the fact that worms speed up when off food to demonstrate that the aggregation mechanism also leads to swarming, and the fact that movement on food is persistent in order to parameterize noise for the model. Otherwise, our model mostly concerns on-food behavior, as 99.7 ± 0.4% (*npr-1*, mean ± S.D.) and 99.8 ± 0.3% (N2, mean ± S.D.) of the worms are on food during the aggregation phase of the experiments.

Either of the two approaches or some combination of them is needed to show convincingly that the postulated behaviors are indeed observed and responsible for the observed population self-organization.

As described above, we have kept the phenomenological model construction and Bayesian parameter inference approaches, but have now added substantial quantitative comparisons between experimental and theoretical outcomes to better illustrate the utility and limitations of our model.

2) The authors discuss possible neuronal mechanisms underlying their behavioural rules. Ideally, it would be good to apply or at least suggest possible experimental tests of these mechanisms, in particular of the proposed important medium-range taxis between neighbouring worms. It is suggested that this might be O2 tension mediated: how might this be tested? They do provide convincing experiments that exclude any significant contribution of ascaroside pheromones to swarming.

We have now suggested several experiments to test the neuronal mechanisms underlying our core behavioral components (Discussion, third paragraph). To elucidate the role of a shallow oxygen gradient in taxis, we suggest introducing mutations leading to aerobic metabolism deficiencies into *npr-1* mutants. Such mutants would still be able to sense ambient oxygen, but are expected to produce an even weaker oxygen gradient in an aggregate. A quantitative comparison of the resulting phenotype to model predictions (e.g. with reduced taxis and/or modified rates of density-dependent reversal and speed switching) would be informative. Additionally, one may perform aerotaxis and oxygen-shift experiments (using a much smaller oxygen gradient than previous works have used) to seek evidence for the ability of worms to sense and move towards environments where oxygen levels are only slightly below ambient concentrations. We also discuss the possibility of perturbing reversals and speed-switching rates using mutations.

Other comments:- The work also mainly compares behaviour in the standard lab strain N2, which is 'hypo-social', to behaviour in a 'hyper-social' npr-1 mutant of N2. Neither of these exactly corresponds to the natural state of almost all wild *C. elegans*, so the analysis is somewhat artificial; this might be more explicitly admitted.

We have now added relevant wording to emphasize that *npr-1* and N2 worms show extreme aggregation phenotypes (Discussion, final paragraph).

- The authors state that to their knowledge "this swarming phenotype has not been reported in *C. elegans* previously", which betrays ignorance. For example, the following observation in Hodgkin and Barnes (2002, PMID: 1684664): "When a population has consumed almost all the bacteria, the worms… form a swarming mass that moves as a wave across the remaining bacterial lawn, and then disperses when all the bacteria have been consumed." This indicates that N2 swarms under appropriate conditions, perhaps of higher bacterial food density. The authors' experiments were performed only on "thin, even bacterial lawns".

We have now included the fact that N2 swarms under appropriate conditions in our text (Results, first paragraph). We have also modified our text in appropriate places to clarify that swarming in this paper refers to that of *npr-1* mutants.

- They also state that "it is still unclear whether aggregation and swarming have a function in the wild". Swarming is almost certainly naturally advantageous in predation on pathogenic or repellent bacteria, as a result of the 'wolfpack' effect that has been studied in myxobacterial swarms, to which some reference should be made. This reviewer, and others working on interactions between *C. elegans* and soil bacteria, have often observed N2 swarming at the edge of unpalatable bacterial lawns. These swarms appear to allow the worms to overcome bacterial defences by collective action of digestive enzymes or other secretions.

We thank the reviewers for sharing these insights and have now included them (Discussion, sixth paragraph). While swarming may be likely to have an ecological role, without a better understanding of the fitness effects and indeed whether it occurs in nature, we would prefer to remain cautious about the function of aggregation and swarming.

- Do worms in the experiments move in purely 2D, i.e. never crawl on top of one another and never overlap? Is this true for the agents in the simulations?

In our experiments, worms move in 2D on agar and rarely burrow. When multiple worms are interacting, they can be observed to crawl over each other, whilst also restricting each other’s movement. This phenomenology does not perfectly correspond to full volume exclusion or free overlap, and thus is difficult to implement in simulations. We chose not to enforce volume exclusion as we suspect the degree of overlap between worms in aggregates to be considerable. Thus, the worms can also overlap in the model, although with taxis direct overlap is discouraged – the direction of taxis reverses (becomes repulsive) if worms overlap, which was chosen to reflect some extent of physical hindrance. We had originally also tried simulations with volume exclusion, but had chosen not to show these as it was not required to achieve aggregation and the aggregates seemed more spread out than in experiments. We now show an example of this in Figure 6—figure supplement 5B.

- Better justification of the level of complexity chosen to model worms with agent-based approach is necessary. Why would simple "vector particle" models fail? How important is that each worm is multi-segmented? How important are the elastic properties of the worm? Does the assumption of the propulsive force generation at the head vs throughout the body affects the motion?

Simple vector-particle models would not have a front and back to detect when particles are exiting a cluster, which seems to be an important behavioral trigger. Though this could be fixed by including a memory, a particle leaving an aggregate ceases to be an interaction partner for particles in the aggregate unless there are long-range interactions. We chose agents with a long shape as we were initially looking for purely local (i.e. short range) interactions that lead to clustering. However, we believe this is still close in simplicity to the “vector particle” model, as our simulated worms really are just “vector particles with a tail”, or equivalently, a chain of vector particles.

The multi-segmented representation of the worms in our model may not be essential, and we suspect that equivalent phenomenology could be captured with rigid rods and different inferred parameters, as long as the rods are allowed to overlap. However, rigid rods have the undesirable property that the tail swings out upon turning, which is not observed in worms. Hence, we chose a persistent random walker with a tail as a more plausible representation in our current approach.

In the model presented here, elastic effects play no importance apart from enforcing non-extensibility of the chain-like worms.

The propulsive force is not generated only at the head; rather, each node of the chain is a self-propelled particle. The head node undergoes a persistent random walk, whereas the other nodes follow the line of the body (and all nodes are subject to non-extensibility constraints). These are described in Figure 5A legend and Appendix 1 under “SPP worm model”.

To investigate whether our model still works with simplified agents, we ran further simulations (starting from the posterior mean parameterization for *npr-1*) with fewer nodes per worm (and correspondingly shorter total worm length, because the size of the nodes was not changed to keep the width the same). We found that aggregation was robust to shortening worms up to a point (Figure 6—figure supplement 5A). With 4 nodes per worm aggregation only formed transiently and with fewer worms per aggregate, indicating that the worms need to be longer than this threshold length. This can be explained by the fact that at this worm length, the interaction radii (set to 3 node radii) of the head and tail nodes start to overlap, and in our simulations worms require a difference in contact between head and tail to initiate reversals. Thus, as worms become shorter than *M*=5 nodes, the head and tail nodes increasingly have the same degree of contact within a cluster, and thus more often fail to sense when they are exiting a cluster (while also having a shorter time window in which to initiate a reversal whilst leaving a cluster).

- By comparison with the bacterial literature, the term swarming used by the authors could be somewhat confusing (e.g. by comparison with the widespread phenomenon referred to as bacterial swarming, e.g. PMID: 20694026). It is also probably acceptable to use the term "swarming" in the worm field, but a clear definition should be added in the Introduction.

We have added an operational definition for *C. elegans* swarming (Results, first paragraph).